# Wide Distribution and Intraspecies Diversity in the Pathogenicity of *Calonectria* in Soil from *Eucalyptus* Plantations in Southern Guangxi of China

**DOI:** 10.3390/jof9080802

**Published:** 2023-07-29

**Authors:** Wenxia Wu, Shuaifei Chen

**Affiliations:** Research Institute of Fast-Growing Trees (RIFT), Chinese Academy of Forestry (CAF), Zhanjiang 524022, China; wuwenxia_hainan@126.com

**Keywords:** *Calonectria* leaf blight, *Eucalyptus* disease, fungal pathogen, pathogenicity, species diversity

## Abstract

*Eucalyptus* spp. are extensively cultivated in southern China because of their adaptability and versatile timber production. *Calonectria* leaf blight caused by *Calonectria* species is considered a major threat to *Eucalyptus* trees planted in China. The GuangXi Zhuang Autonomous Region is the provincial region with the largest distribution of *Eucalyptus* plantations in China. The present study aimed to expound the species diversity and pathogenicity of *Calonectria* isolates obtained from the soil of *Eucalyptus* plantations in GuangXi. A total of 188 *Calonectria* isolates were recovered from the soil located close to *Eucalyptus* trees, and the isolates were identified based on the DNA sequence comparisons of the four partial regions of the translation elongation factor 1-alpha (*tef1*), β-tubulin (*tub2*), calmodulin (*cmdA*), and histone H3 (*his3*) genes. The isolates were identified as *Calonectria aconidialis* (74.5%), *C*. *hongkongensis* (21.3%), *C*. *pseudoreteaudii* (2.1%), *C*. *kyotensis* (1.6%), and *C*. *chinensis* (0.5%). The inoculation results indicated that 40 isolates representing five *Calonectria* species were pathogenic to the three *Eucalyptus* genotypes. Two inoculated experiments consistently showed that the longest lesions were produced by the isolates of *C*. *aconidialis*. Some isolates of *C*. *aconidialis*, *C*. *hongkongensis*, and *C*. *kyotensis* produced significantly longer lesions than the positive controls, but not the isolates of *C*. *pseudoreteaudii* or *C*. *chinensis*. These results indicated that *Calonectria* isolated from the soil may pose a threat to *Eucalyptus* plantations. Some *Calonectria* isolates of the same species differed significantly in their virulence in the tested *Eucalyptus* genotypes. The resistance of different *Eucalyptus* genotypes to *Calonectria* isolates within the same species was inconsistent. The inoculation results in this study suggested that many *Calonectria* isolates in each species had different levels of pathogenicity, and many *Eucalyptus* genotypes need to be tested to select disease-resistant *Eucalyptus* genetic materials in the future. The results of the present study enhance our knowledge of species diversity and the potential damage caused by *Calonectria* in the soil of *Eucalyptus* plantations. Our results also provide new insights into the breeding of disease-resistant *Eucalyptus* genotypes for controlling *Calonectria* leaf blight in China in the future.

## 1. Introduction

*Eucalyptus* L’Hér. (Myrtaceae Juss., Myrtales Juss. ex Bercht. and J.Presl), due to its rapid growth, robust adaptability, and broad applications, is extensively planted in tropical and subtropical regions in China [1]. *Eucalyptus* was originally introduced to China in 1890 as an ornamental plant [1]. The area covered by *Eucalyptus* plantations has increased exponentially, from 0.46 million hm^2^ in 1986 to 5.46 million hm^2^ in 2018 [2]. In China, *Eucalyptus* plantations are distributed mainly in GuangXi, GuangDong, YunNan, FuJian, SiChuan, and HaiNan Provinces (or Autonomous Regions). The GuangXi Zhuang Autonomous Region is the provincial region with the largest distribution of *Eucalyptus* plantations in this country [1]. The area of *Eucalyptus* plantations in GuangXi is 2.56 million hm^2^, which is 46.83% of the total area of *Eucalyptus* plantations in the country [3].

Over the past three decades, *Eucalyptus* plantations in China have experienced a significant threat of diseases [4,5,6]. Leaf blight caused by *Calonectria* De Not. species is considered one of the major threats to plantations [5,7,8,9]. *Calonectria* species primarily infect the leaves of the middle and lower parts of *Eucalyptus* trees, resulting in water-soaked spots. Under high temperatures and humidity, the spots gradually develop into extended necrotic areas, eventually causing the whole leaves to become blighted and fall off [5,7,9,10,11]. These species also cause cutting rot, damping-off, stem rot, and leaf rot in *Eucalyptus* nurseries [5,11]. *Eucalyptus* leaf blight caused by *Calonectria* species also occurs in other countries, including Australia, Brazil, India, Indonesia, Malaysia, Thailand, and Vietnam [10,12,13,14,15,16,17].

The genus *Calonectria* includes important pathogens that infect more than 335 plant species, distributed among nearly 100 plant families. These plants include forestry, agricultural, and horticultural crops [11,18,19]. In forestry, *Calonectria* species mainly attack the families Fabaceae Lindl., Myrtaceae, and Pinaceae Lindl. [11,18].

*Calonectria* species are soil-borne fungi and their microsclerotia can survive in the soil for extended periods [11]. Currently, 137 *Calonectria* species have been discovered worldwide [12,15,16,20,21,22,23]. Among these, 84 species have been isolated from the soil near agricultural crops, plantations, natural forests, and unknown forest types in Asia, Africa, North America, and South America [9,12,15,16,20,22,23,24,25,26,27].

Several *Calonectria* species isolated from blighted *Eucalyptus* leaves and soil in *Eucalyptus* plantations in China were pathogenic to the tested *Eucalyptus* genotypes [7,8,9,28,29]. Some of these species were acquired from diseased *Eucalyptus* tissues (leaves and branches) and soil close to these trees. In the present study, soil samples were obtained from *Eucalyptus* plantations in GuangXi. The purposes of this study were to (i) expound the species diversity of *Calonectria* isolated from these soil samples, and (ii) clarify the pathogenicity of *Calonectria* species on different *Eucalyptus* genotypes.

## 2. Materials and Methods

### 2.1. Sample Site, Collection, and Fungal Isolation

Soil samples were collected from *Eucalyptus* plantations between July and August 2019 in GuangXi, southern China. These plantations were located at seven sampling sites across four regions, BeiHai, QinZhou, FangchengGang, and ChongZuo Region (Figure 1, Table 1). The soil in the 3–5-year-old *Eucalyptus* plantations was relatively moist with thick layers of leaf litter. The upper 0–20 cm of the soil was extracted by removing the thick layers of leaf litter. Fifty-three to sixty-nine soil samples were randomly collected from each sample site (Table 1). The soil samples were first placed in plastic bags to maintain humidity and temperature and then transferred to a laboratory for fungal isolation and further molecular studies.

To induce *Calonectria* isolates, distilled water was utilized to moisten the soil samples in plastic cups. *Medicago sativa* L. (alfalfa) seeds were surface disinfested in 75% ethanol for 30 s and washed with distilled water. They were then placed on the surface of the moistened soil in plastic cups, as described by Crous [11]. The sampling cups with soil and alfalfa seeds were incubated at 25 °C under 12 h of daylight and 12 h of darkness. After 7 d, the sampling cups with soil and germinating alfalfa seedlings were observed under a dissection microscope. *Calonectria* isolates were distinguished from other fungi based on the typical morphological characteristics of conidiophores, macroconidia, and vesicles [11,18,30]. A single conidium was transferred from the conidiophores of *Calonectria* to a 2% (*v*/*v*) malt extract agar (MEA) (20 g of malt extract powder and 20 g of agar powder per liter of water) using sterile needles under a stereoscopic microscope. For each soil sample, a culture of one morphologically similar *Calonectria* isolate was retained for further studies. The obtained cultures were deposited in the culture collection (CSF) located at the Research Institute of Fast-growing Trees (RIFT) of the Chinese Academy of Forestry (CAF) in ZhanJiang, GuangDong Province, China.

### 2.2. DNA Extraction, PCR Amplification, and Sequencing

The DNA was extracted after the isolates were grown on MEA for 7–10 days. Mycelia were carefully scraped from the surface of the MEA culture medium using a sterilized scalpel and transferred to a 2 mL Eppendorf tube. Total genomic DNA was extracted according to “Extraction method 5: grinding and CTAB” protocols described by van Burik et al. [31]. The extracted DNA was dissolved in 30 µL of TE buffer (1 M Tris-HCl and 0.5 M EDTA, pH 8.0), and then 3 µL of RNase (10 mg/mL) was added at 37 °C for 1 h to degrade the RNA. In the final step, a Nano-Drop 2000 spectrometer (Thermo Fisher Scientific, Waltham, MA, USA) was used to measure the DNA concentration.

Consistent with previous studies, the use of four loci, partial gene regions of translation elongation factor 1-alpha (*tef1*), β-tubulin (*tub2*), calmodulin (*cmdA*), and histone H3 (*his3*), was successful in identifying *Calonectria* species [5,12,21,22,32,33,34]. These four partial gene regions were amplified using specific primer pairs: EF1-728F/EF2 for the *tef1* gene region; fRpb2-5F/fRpb2-7cR or T1/CYLTUB1R for the *tub2* gene region; CAL-228F/CAL-2Rd for the *cmdA* gene region; and CYLH3F/CYLH3R for the *his3* gene region [22,24,35]. The PCR reaction mixtures contained 17.5 μL of TopTaq ^TM^ master mix, 1 μL of each primer (10 mM), 2 μL of the DNA sample, and RNase-free H_2_O adjusted to a final volume of 35 μL. The amplification was conducted according to the conditions described by Liu et al. [22].

All the PCR products were sequenced in both the forward and reverse directions of each primer pair at the Beijing Genomics Institute, GuangZhou, China. The sequences were manually edited using MEGA v. 6.0 software [36] and then submitted to GenBank (https://www.ncbi.nlm.nih.gov, accessed on 8 March 2023).

### 2.3. Phylogenetic Analyses

To preliminarily identify the isolates, a standard nucleotide BLAST search was performed using the *tef1*, *tub2*, *cmdA*, and *his3* sequences. The sequences of the available species in the relevant species complexes were downloaded from NCBI for sequence comparisons and phylogenetic analyses. The alignment of sequences for each of the *tef1*, *tub2*, *cmdA*, and *his3* gene regions, as well as the combination of these four gene regions, was performed online using MAFFT v. 7 (http://mafft.cbrc.jp/alignment/server/, accessed on 8 March 2023) with alignment strategy FFT-NS-i (slow; interactive refinement method) [37]. The manual sequence adjustment was performed using MEGA v. 7 software [38].

The maximum likelihood (ML) and Bayesian inference (BI) methods were used for the phylogenetic analysis of the sequence datasets of each of the four gene regions and the combination of these regions. The optimal models of the five sequence datasets for BI analyses were determined using the jModelTest v. 2.1.5 [39]. ML analyses were performed using RaxML v. 8.2.12 [40] on the CIPRES Science Gateway v. 3.3, with the default GTR substitution matrix and 1000 bootstrap runs. The software MrBayes v. 3.2.7 [41] was used for BI analyses with CIPRES Science Gateway v. 3.3. Four Markov chain Monte Carlo (MCMC) chains were executed from a random starting tree for five million generations, and the trees were sampled every 100th generation. The first 25% of the trees were discarded as burn-in, and the rest of the trees were used to confirm the posterior probabilities. Phylogenetic trees were viewed using MEGA v. 7 [38] and FigTree v 1.4.2 for ML and BI trees, respectively. The sequence data for CBS 109167 and CBS 109168 (*Curvicladiella cignea* Decock and Crous) were treated as outgroups [22].

### 2.4. Pathogenicity Tests

Representative isolates of each *Calonectria* species identified in this study were selected for inoculation trials. Three *Eucalyptus* genotypes were selected for inoculation, *E*. *urophylla* S. T. Blake × *E*. *tereticornis* Sm. hybrid genotype CEPT1900 and *E*. *urophylla* × *E*. *grandis* W. Hill hybrid genotypes CEPT1901 and CEPT1902. All inoculated seedlings were similar in size, 3 months old, and approximately 40 cm in height.

Inoculation with mycelial plugs was performed as described by Wu and Chen [9]. For each *Eucalyptus* genotype, 10 mycelial plugs (5 mm diameter) from 7-day-old MEA cultures of each isolate were inoculated on the abaxial surface of the unwounded leaves of three *Eucalyptus* seedlings. Ten leaves from three different *Eucalyptus* seedlings treated with sterile MEA plugs were used as negative controls. The highly pathogenic *Calonectria pseudoreteaudii* L. Lombard, M.J. Wingf. and Crous, isolate CSF13317 of two *Eucalyptus* hybrid genotypes, *E. urophylla × E. grandis* genotype CEPT1878 and *E. urophylla × E. tereticornis* genotype CEPT1879, as confirmed in a previous study, was used as a positive control [29]. To ensure sufficient humidity for infection development, all *Eucalyptus* seedlings were placed in moist plastic chambers and maintained under stable climatic conditions (temperature 25–26 °C; humidity 60–70%) for three days. The plastic chambers were removed after three days. To measure the lesion length of each leaf, two diameter measurements of each lesion perpendicular to each other were conducted for each leaf, and the average lesion diameter was computed. The entire experiment was repeated using an identical methodology. The inoculations were conducted in July 2022 at the South China Experimental Nursery (SCEN), located in ZhanJiang, GuangDong Province, China.

To verify Koch’s postulates, re-isolations were conducted. Small pieces of discolored leaf tissue (approximately 0.04 cm^2^) from the periphery of the generated lesions were cut and placed on a 2% MEA at room temperature. For each inoculated isolate, four leaves of each *Eucalyptus* genotype were randomly selected, and all the leaves inoculated as positive and negative controls were re-isolated. The re-isolated fungi were identified and confirmed based on the morphological characteristics and disease symptoms exhibited by the leaves with the original fungi. Statistical analyses were performed by one-way analysis of variance (ANOVA) using SPSS Statistics 22 software (IBM Corp., Armonk, NY, USA).

## 3. Results

### 3.1. Sample Collection and Fungal Isolation

A total of 428 soil samples were collected from seven sampling sites (A to G) in four regions in GuangXi (Figure 1, Table 1). The fungi with branched conidiophores producing cylindrical conidia and with stipe extensions terminating in a vesicle with a characteristic shape were grouped as *Calonectria*. A total of 188 soil samples, which accounted for 43.9% of all sampled soil samples, were positive for *Calonectria* isolates with branched conidiophores, cylindrical macroconidia, and sphaeropedunculate or clavate vesicles. For each sample, a single conidium culture was isolated from white masses of conidiophores with typical morphological characteristics of *Calonectria* species. In total, 188 *Calonectria* isolates were obtained from 188 soil samples. The percentage of soil samples that yielded *Calonectria* ranged from 5.0% to 62.3% at the seven sampling sites (Table 1).

### 3.2. Sequencing

DNA extraction and sequence comparisons of all 188 *Calonectria* isolates were performed (Table 2). The *tef1*, *tub2*, *cmdA*, and *his3* gene regions of all 188 isolates were amplified. The obtained sequence fragments for the *tef1*, *tub2*, *cmdA*, and *his3* gene regions were approximately 520, 600, 690, and 460 bp, respectively. Based on the sequences of the *tef1*, *tub2*, *cmdA*, and *his3* loci, the genotypes of all 188 sequenced isolates were determined. A total of 32 genotypes were identified (Table 2).

### 3.3. Phylogenetic Analyses

For the 188 isolates sequenced in this study, one to two isolates of each genotype determined by *tef1*, *tub2*, *cmdA*, and *his3* sequences were selected for phylogenetic analyses. A total of 47 representative isolates representing 32 genotypes were selected (Table 2). The sequences of 69 isolates presenting 40 published *Calonectria* species closely related to the *Calonectria* isolates obtained in the present study were downloaded from GenBank and used for phylogenetic analyses based on four individual gene regions and the combination of those regions (Table 3).

For BI phylogenetic analyses of each dataset, GTR+I, TPM2uf+I+G, TIM1+G, TPM2uf+I+G, and GTR+I+G models were selected for *tef1*, *tub2*, *cmdA*, *his3*, and the combination of those regions, respectively. The overall topologies generated from the ML analyses and the BI analyses for each dataset were similar. The ML tree with bootstrap support values and the posterior probabilities obtained from BI are presented in Figure 2 and Appendix A.

The 47 *Calonectria* isolates were divided into five groups (Groups A to E) based on *tef1*, *tub2*, *cmdA*, *his3*, and combined *tef1*/*tub2*/*cmdA*/*his3* analyses (Figure 2 and Appendix A). The phylogenetic analyses showed that the isolates in Groups A, B, C, and D belong to the *C*. *kyotensis* species complex, while the isolates in Group E belong to the *C*. *reteaudii* species complex.

The isolates in Group A represented 19 genotypes based on the sequences of four gene regions (Table 2). The phylogenetic analyses showed that these isolates were grouped with *Calonectria aconidialis* L. Lombard, Crous and S.F. Chen based on the *tef1*, *cmdA*, and *his3* trees (Appendix A). In the *tub2* tree, the isolates were clustered directly with or most closely to *C*. *aconidialis*, *Calonectria asiatica* Crous and Hywel-Jones, and *Calonectria uniseptate* Gerlach (Appendix A), and were grouped with *C*. *aconidialis* according to the combined *tef1*/*tub2*/*cmdA*/*his3* tree (Figure 2). Therefore, the isolates in Group A were identified as *C*. *aconidialis*. The isolates in Group B represented one genotype (Table 2). These isolates were clustered with *Calonectria kyotensis Terash.* in the *tef1*, *tub2*, and *his3* trees (Appendix A), and were clustered directly with or most closely to *C*. *kyotensis* and *C*. *uniseptate* in the *cmdA* tree (Appendix A). According to the combined *tef1/tub2/cmdA/his3* tree, these isolates were grouped with *C*. *kyotensis* (Figure 2), and therefore isolates in Group B were identified as *C*. *kyotensis*. The isolates in Group C represented 10 genotypes (Table 2) and were clustered with *Calonectria hongkongensis* Crous in the *tef1*, *tub2*, *cmdA*, and *his3* trees and the four-gene combined phylogenetic tree (Figure 2 and Appendix A). The isolates in Group C were identified as *C*. *hongkongensis*. The isolate in Group D represented one genotype (Table 2). This isolate was clustered with *Calonectria chinensis* (Crous) L. Lombard, M.J. Wingf. and Crous in the *cmdA* and *his3* trees (Appendix A). The isolate was clustered directly with or most closely to *C*. *chinensis* in the *tef1* and *tub2* trees (Appendix A). The isolate was clustered with *C*. *chinensis* based on the combined *tef1*/*tub2*/*cmdA*/*his3* tree (Figure 2). Consequently, the isolate was identified as *C*. *chinensis*.

The isolates in Group E represented one genotype (Table 2). These isolates were clustered with *C*. *pseudoreteaudii* in the *tef1*, *tub2*, and *his3* trees (Appendix A). These isolates were grouped with *C*. *pseudoreteaudii* and *Calonectria reteaudii* (Bugnic.) C. Booth in the *cmdA* tree (Appendix A). According to the combined *tef1*/*tub2*/*cmdA*/*his3* tree, these isolates were grouped with *C*. *pseudoreteaudii* (Figure 2). Therefore, the isolates in Group E were identified as *C*. *pseudoreteaudii*.

### 3.4. Diversity and Distribution of Calonectria Species

Based on the sequence comparisons of the four gene region sequences, the 188 *Calonectria* isolates were identified as five species, *C*. *aconidialis* (74.5%), *C*. *hongkongensis* (21.3%), *C*. *pseudoreteaudii* (2.1%), *C*. *kyotensis* (1.6%), and *C*. *chinensis* (0.5%) (Figure 3). *Calonectria hongkongensis* was isolated from all seven sampling sites (Sites A to G) (Table 2). *Calonectria aconidialis* was isolated from six sampling sites (Sites A to F) (Table 2). *Calonectria pseudoreteaudii* was detected at sites C, E, and G (Table 2). *Calonectria kyotensis* and *C*. *chinensis* were found only at sites F and G, respectively (Table 2).

### 3.5. Pathogenicity Tests

Forty isolates representing the five *Calonectria* species, *C*. *aconidialis* (21 isolates), *C*. *hongkongensis* (12 isolates), *C*. *kyotensis* (three isolates), *C*. *pseudoreteaudii* (three isolates), and *C*. *chinensis* (one isolate), were used for pathogenicity tests on the leaves of three *Eucalyptus* genotypes (Table 2, Figure 4 and Figure 5). All 40 isolates and the positive control produced disease spots and lesions on the leaves of the inoculated seedlings. No disease symptoms were observed in the leaves of the negative control seedlings (Figure 4 and Figure 5). *Calonectria* species with the same morphological characteristics as the originally inoculated fungi were successfully re-isolated from the diseased tissues of the inoculated leaves. No *Calonectria* isolates were re-isolated from the leaves of the negative control seedlings. Thus, Koch’s postulates were fulfilled. Two pathogenicity tests were performed, and ANOVA showed that the two pathogenicity tests were significantly different (*p* < 0.05). Consequently, the data from each experiment were analyzed separately.

The results of the pathogenicity tests showed that some isolates of *C*. *aconidialis*, *C*. *hongkongensis*, and *C*. *kyotensis* generated significantly longer lesions than the positive control on each of the three *Eucalyptus* genotypes in both experiments (*p* < 0.05). For example, *C*. *aconidialis* isolates (CSF16507, CSF16520, CSF16557, CSF16582, CSF16648, CSF16693, CSF16706, CSF17110, CSF17130, and CSF17142), *C*. *hongkongensis* isolate CSF17125, and *C*. *kyotensis* isolate CSF16776 produced significantly longer lesions than positive control isolate CSF13317 (*C*. *pseudoreteaudii*) on the three *Eucalyptus* genotypes in both experiments (Figure 4 and Figure 5). 

Significant differences in pathogenicity were also observed among isolates of the same *Calonectria* species in both experiments (*p* < 0.05). For example, *C*. *aconidialis* isolate CSF16706 produced significantly longer lesions (*p* < 0.05) than the other *C*. *aconidialis* isolates (CSF16470, CSF16507, CSF16522, CSF16527, CSF16582, CSF16584, CSF16599, CSF16609, CSF16643, CSF16653, CSF16675, CSF16718, and CSF16742) on each of the three *Eucalyptus* genotypes in both experiments. *Calonectria hongkongensis* isolate CSF16754 produced significantly longer lesions (*p* < 0.05) than the other *C*. *hongkongensis* isolates (CSF16726, CSF16737, CSF16781, CSF16786, and CSF16823). *Calonectria kyotensis* isolate (CSF16776 produced significantly longer lesions (*p* < 0.05) than the other *C*. *kyotensis* isolates CSF16724 and CSF16801) (Figure 4 and Figure 5).

The pathogenicities of the same genotype determined by the sequences of *tef1*, *tub2*, *cmdA*, and *his3* loci of the same *Calonectria* species were significantly different in the three *Eucalyptus* genotypes (*p* < 0.05). For example, *C*. *aconidialis* isolates (genotype: AAAA) CSF17130 and CSF17142 produced significantly longer lesions (*p* < 0.05) than those caused by CSF17110 in experiment one (Table 2, Figure 4), and CSF17110 produced significantly longer lesions (*p* < 0.05) than CSF17142 in experiment two (Table 2, Figure 5). *Calonectria hongkongensis* (genotype: AAAA) CSF17125 produced significantly longer lesions (*p* < 0.05) than CSF16726 and CSF16756 in experiment one (Table 2, Figure 4), and *C*. *kyotensis* (genotype: AAAA) isolate CSF16776 produced significantly longer lesions (*p* < 0.05) than CSF16724 and CSF16801 in both experiments (Table 2, Figure 4 and Figure 5).

The overall data showed that *Eucalyptus* genotypes CEPT1900 and CEPT1901 were relatively more tolerant than CEPT1902 to the *Calonectria* isolates tested in this study. The majority of the tested *Calonectria* isolates generated longer lesions on genotype CEPT1902 than on CEPT1900 and CEPT1901 in both experiments, except for *C*. *aconidialis* isolates (CSF16599, CSF16648, and CSF16675), *C*. *hongkongensis* isolates (CSF16463, CSF16731, and CSF16754), and *C*. *kyotensis* isolate CSF16776 in experiment one (Figure 4), and *C*. *aconidialis* isolates (CSF16520, CSF16718, and CSF17110), *C*. *hongkongensis* isolates (CSF16726, CSF16731, CSF16754, CSF16756, and CSF16823), *C*. *pseudoreteaudii* isolate CSF16826, *C*. *kyotensis* isolate CSF16801, and *C*. *chinensis* isolate CSF16829 in experiment two (Figure 5).

The resistance of different *Eucalyptus* genotypes to *Calonectria* isolates within the same species was inconsistent. For example, *Eucalyptus* genotypes CEPT1900 and CEPT1901 were significantly more tolerant than genotype CEPT1902 in both experiments (*p* < 0.05) to *C*. *aconidialis* isolates (CSF16470, CSF16522, CSF16527, CSF16742, and CSF17130), and *C*. *hongkongensis* isolate CSF17118 was significantly more tolerant than genotype CEPT1902 in both experiments (*p* < 0.05). *Eucalyptus* genotype CEPT1901 was significantly more tolerant to *C*. *aconidialis* isolate CSF16599 than to the other two *Eucalyptus* genotypes in both experiments (*p* < 0.05) (Figure 4 and Figure 5).

## 4. Discussion

In this study, 428 soil samples were collected from seven *Eucalyptus* plantations in multiple regions of GuangXi in southern China. Based on their morphological characteristics, 188 *Calonectria* isolates were obtained. Of these, 188 isolates were identified based on multi-gene phylogenetic inferences. These isolates were identified as *C*. *aconidialis*, *C*. *hongkongensis*, *C*. *pseudoreteaudii*, *C*. *kyotensis*, and *C*. *chinensis*. Pathogenicity tests indicated that all five *Calonectria* species were pathogenic among the three tested *Eucalyptus* genotypes.

This study showed that *Calonectria* fungi in the *C*. *kyotensis* species complex were widely distributed in the soil of *Eucalyptus* plantations in southern China. *Calonectria* fungi were isolated from 43.9% of the soil samples. Except for *C*. *pseudoreteaudii*, which reside in the *C*. *reteaudii* species complex, the other four species resided in the *C*. *kyotensis* species complex. The four species in the *C*. *kyotensis* species complex accounted for 97.9% of all the isolates obtained in this study. This is consistent with the results of previous studies showing that *Calonectria* species in the *C*. *kyotensis* species complex, especially *C*. *aconidialis*, *C*. *hongkongensis*, and *C*. *kyotensis*, are the dominant species distributed in the soil of *Eucalyptus* plantations in southern China [9,23,24,25]. In addition to soil isolation, *C. aconidialis*, *C. kyotensis*, *C*. *hongkongensis,* and *C. chinensis* were also occasionally isolated from diseased *Eucalyptus* tissues [8,28,55]. *Calonectria hongkongensis* also caused fruit rot in rambutan (*Nephelium lappaceum* L.) in Puerto Rico [57]. The *Calonectria* species in the *C. kyotensis* species complex isolated from the soil in this study can cause disease in *Eucalyptus* trees.

In this study, one species in the *C*. *reteaudii* species complex, *C*. *pseudoreteaudii*, was isolated from the soil of three *Eucalyptus* plantation sites. This fungus has been extensively isolated from diseased *Eucalyptus* tissues (leaves and branches) in plantations in FuJian, GuangXi, GuangDong, and HaiNan Provinces in southern China [5,7,8,9,21,43,58]. *Calonectria pseudoreteaudii* is considered one of the key causal agents of *Eucalyptus* leaf blight in southern China. Except for *Eucalyptus* trees, *C. pseudoreteaudii* caused leaf spots in *Macadamia* F. Muell. sp. in China and Laos [59,60] and caused leaf spot and stem blight in *Vaccinium corymbosum* L. in China [61]. The results of this and previous studies suggested that *C. pseudoreteaudii* is an important pathogen in many plant species with a wide geographic distribution.

The inoculation results indicated that isolates of *C*. *aconidialis*, *C*. *hongkongensis*, and *C*. *kyotensis* produced significantly longer lesions than those of the positive control on the three *Eucalyptus* genotypes. These results highlighted that *Calonectria* species dominantly distributed in the soil were potential threats to *Eucalyptus* plantations in southern China.

One of the most effective measures to control *Eucalyptus* leaf blight caused by *Calonectria* species is selecting disease-resistant *Eucalyptus* genotypes. *Eucalyptus* genotypes resistant to *Eucalyptus* leaf blight have been selected in Australia, Brazil, China, India, and South Africa [62,63,64,65,66,67].

The pathogenicity tests in this study indicated that some *Calonectria* isolates within the same species were significantly different in their virulence from the tested *Eucalyptus* genotypes. The resistance of different *Eucalyptus* genotypes to *Calonectria* isolates within the same species was inconsistent. This is consistent with the results of previous studies [9,29]. Variations in plant pathogen intra-species pathogenicity and differences in plant pathogen resistance are common in some pathogens and plants [68,69,70]. This is the result of the evolution of both pathogenicity and virulence of plant pathogens, pathogen and plant genetic regulation, plant pathogen co-evolution, and other factors [71,72,73]. The results of this and previous studies suggested that in the process of selecting disease-resistant *Eucalyptus* genotypes, many isolates of each *Calonectria* species with different pathogenicities and many *Eucalyptus* genotypes should be tested.

## Figures and Tables

**Figure 1 jof-09-00802-f001:**
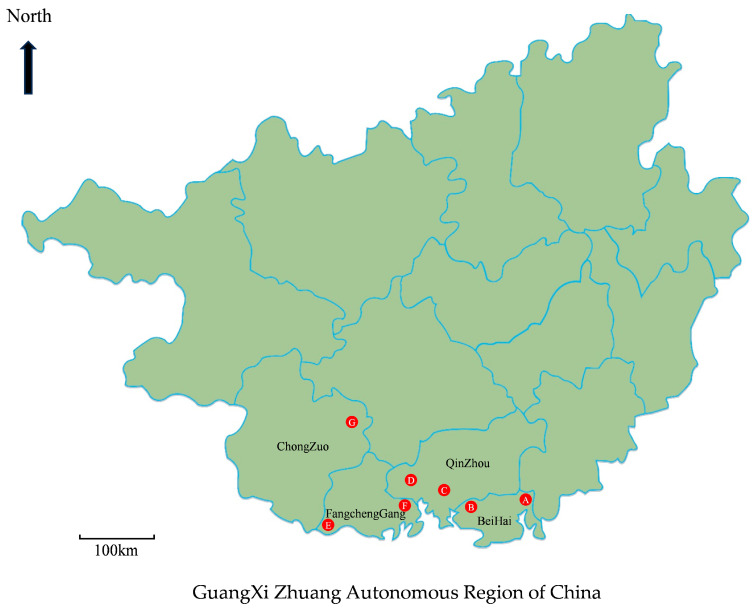
Map of GuangXi Zhuang Autonomous Region showing sampling sites in this study. The seven sampling sites are indicated as letters A to G.

**Figure 2 jof-09-00802-f002:**
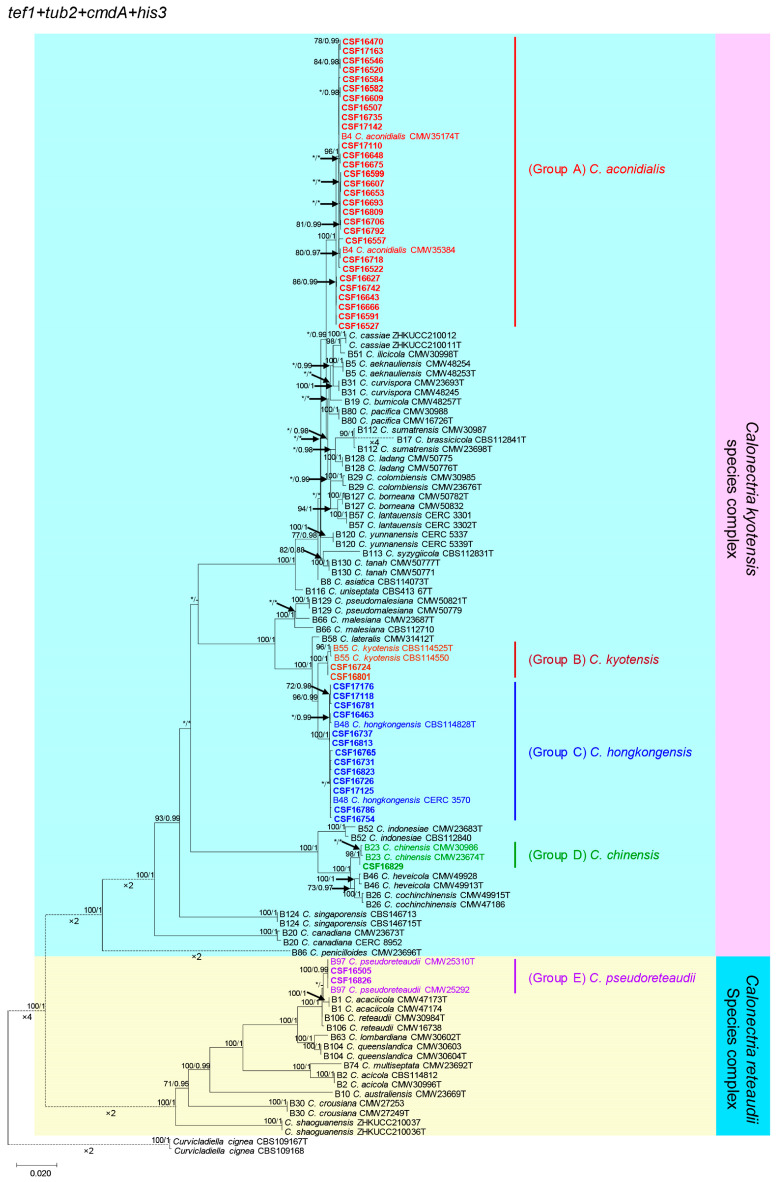
Phylogenetic tree of *Calonectria* species based on maximum likelihood (ML) analyses of a combined DNA dataset of *tef1*, *tub2*, *cmdA*, and *his3* gene sequences. Bootstrap support value ≥ 70% for ML and posterior probabilities values ≥ 0.95 for Bayesian inference (BI) analyses are presented above the branches as follows: ML/BI. Bootstrap values < 70% or probabilities values < 0.95 are marked with “*”, and absent analysis values are marked with “-” Ex-type isolates are marked with “T”. Isolates sequenced in this study are highlighted in bold and shown in color. Two isolates of *Curvicladiella cignea* (CBS 109167 and CBS 109168) were used as outgroups.

**Figure 3 jof-09-00802-f003:**
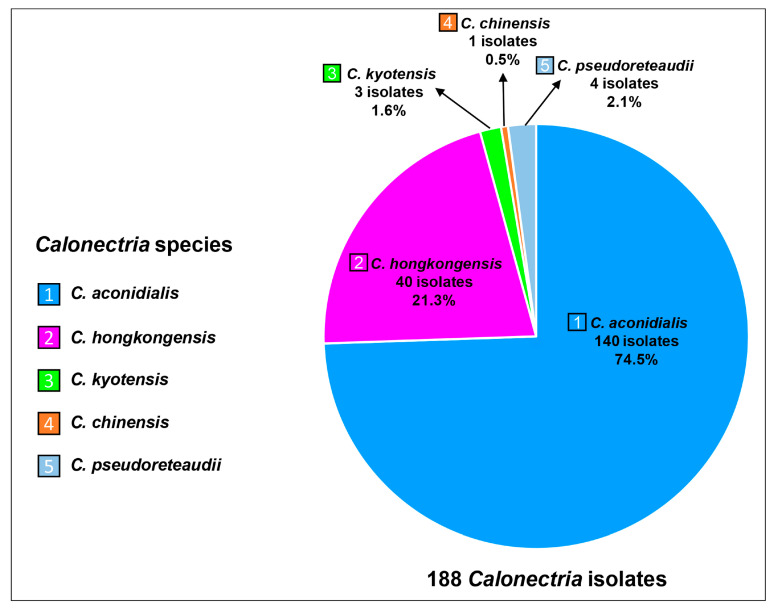
The isolate number and percentage of each *Calonectria* species in the GuangXi Zhuang Autonomous Region. Different species are indicated by numbers with different colors.

**Figure 4 jof-09-00802-f004:**
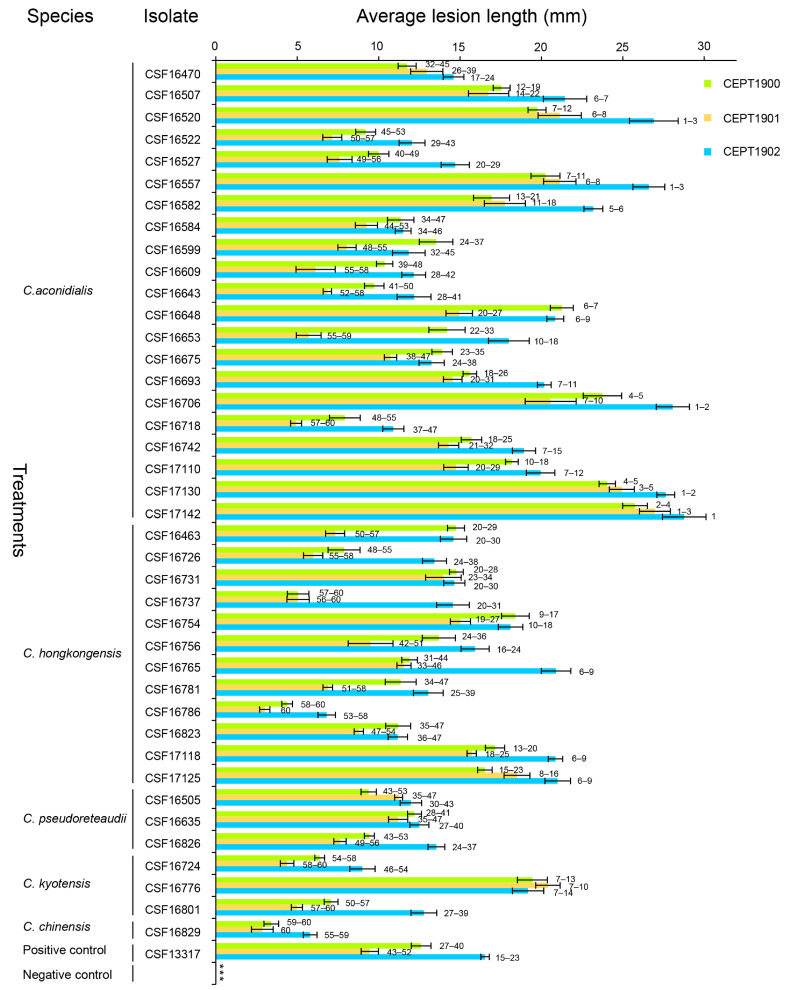
The pathogenicity results of experiment one. Column chart indicating the average lesion length (mm) on leaves resulting from inoculation trials of three *Eucalyptus* hybrid genotypes inoculated with 40 isolates of five *Calonectria* species and positive and negative controls. Horizontal bars represent the standard error of the means. Different numbers on the right of the bars indicate treatment means that were significantly different (*p* = 0.05). The “***” represents no lesions produced by the negative controls.

**Figure 5 jof-09-00802-f005:**
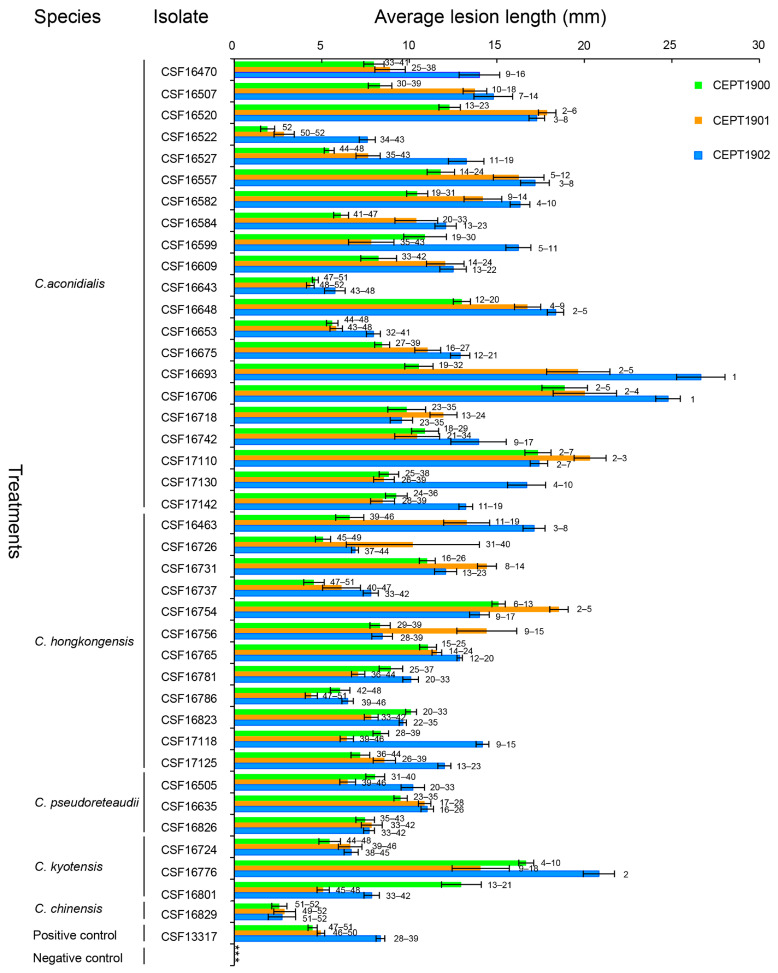
The pathogenicity results of experiment two. Column chart indicating the average lesion length (mm) on leaves resulting from inoculation trials of three *Eucalyptus* hybrid genotypes inoculated with 40 isolates of five *Calonectria* species and positive and negative controls. Horizontal bars represent the standard error of the means. Different numbers on the right of the bars indicate treatment means that were significantly different (*p* = 0.05). The “***” represents no lesions produced by the negative controls.

**Table 1 jof-09-00802-t001:** Soil samples and recovered *Calonectria* isolates from *Eucalyptus* plantations in this study.

Site No.	Region	Location	GPS Coordinate	Number of Soil Samples	Number of Soil Samples with *Calonectria*	Percentage of Soil Samples with *Calonectria*
A	BeiHai	LongJiang Village, BaiSha Town, HePu County, BeiHai Region	21°46′7.0464″ N, 109°39′27.5256″ E	62	35	56.5%
B	BeiHai	DongXin Village, ShiWan Town, HePu County, BeiHai Region	21°47′28.3848″ N, 109°13′14.2608″ E	60	3	5.0%
C	QinZhou	ChangeDong Village, NaLi Town, QinNan District, QinZhou Region	21°51′11.83″ N, 108°51′13.18″ E	60	36	60.0%
D	QinZhou	TunNan Village, HuangwuTun Town, QinNan District, QinZhou Region	21°58′22.41″ N, 108°29′16.78″ E	53	33	62.3%
E	FangchengGang	SongBai Village, DongXing Town, FangchengGang Region	21°34′53.37″ N, 108°04′20.53″ E	69	39	56.5%
F	FangchengGang	NaYong Village, FangchengGang District, FangchengGang Region	21°52′49.60″ N, 108°17′46.06″ E	63	38	60.3%
G	ChongZuo	NaPo Village, FuSui County, ChongZuo Region	22°35′06.39″ N, 107°57′19.75″ E	61	4	6.6%
Total				428	188	43.9%

**Table 2 jof-09-00802-t002:** Isolates obtained in this study used for phylogenetic analyses and pathogenicity tests.

Species	Isolate No. ^a,b,c^	Genotype ^d^	Sampling Sites	Sample No.	Host	Collector	GenBank Accession No. ^e^			
							*tef1*	*tub2*	*cmdA*	*his3*
*C. aconidialis*	CSF16467	AAAA	B	20190704-2-(26)	soil under 3-year-old *E. urophylla × E. grandis*	S.F. Chen, Q.C. Wang, W.X. Wu, Y.X. Zheng & L.F. Liu	OR261297	OR261485	OR261673	OR261861
*C. aconidialis*	CSF16470 ^b,c^	AHAA	C	20190704-3-(1)	soil under 3-year-old *E. urophylla × E. grandis*	S.F. Chen, Q.C. Wang, W.X. Wu, Y.X. Zheng & L.F. Liu	OR261298	OR261486	OR261674	OR261862
*C. aconidialis*	CSF16473	AFBA	C	20190704-3-(2)	soil under 3-year-old *E. urophylla × E. grandis*	S.F. Chen, Q.C. Wang, W.X. Wu, Y.X. Zheng & L.F. Liu	OR261299	OR261487	OR261675	OR261863
*C. aconidialis*	CSF16477	AAAA	C	20190704-3-(4)	soil under 3-year-old *E. urophylla × E. grandis*	S.F. Chen, Q.C. Wang, W.X. Wu, Y.X. Zheng & L.F. Liu	OR261300	OR261488	OR261676	OR261864
*C. aconidialis*	CSF16479	AFBA	C	20190704-3-(5)	soil under 3-year-old *E. urophylla × E. grandis*	S.F. Chen, Q.C. Wang, W.X. Wu, Y.X. Zheng & L.F. Liu	OR261301	OR261489	OR261677	OR261865
*C. aconidialis*	CSF16481	AAAA	C	20190704-3-(6)	soil under 3-year-old *E. urophylla × E. grandis*	S.F. Chen, Q.C. Wang, W.X. Wu, Y.X. Zheng & L.F. Liu	OR261302	OR261490	OR261678	OR261866
*C. aconidialis*	CSF16484	AAAA	C	20190704-3-(11)	soil under 3-year-old *E. urophylla × E. grandis*	S.F. Chen, Q.C. Wang, W.X. Wu, Y.X. Zheng & L.F. Liu	OR261303	OR261491	OR261679	OR261867
*C. aconidialis*	CSF16488	AAAA	C	20190704-3-(13)	soil under 3-year-old *E. urophylla × E. grandis*	S.F. Chen, Q.C. Wang, W.X. Wu, Y.X. Zheng & L.F. Liu	OR261304	OR261492	OR261680	OR261868
*C. aconidialis*	CSF16490	AHAA	C	20190704-3-(14)	soil under 3-year-old *E. urophylla × E. grandis*	S.F. Chen, Q.C. Wang, W.X. Wu, Y.X. Zheng & L.F. Liu	OR261305	OR261493	OR261681	OR261869
*C. aconidialis*	CSF16493	AAAA	C	20190704-3-(15)	soil under 3-year-old *E. urophylla × E. grandis*	S.F. Chen, Q.C. Wang, W.X. Wu, Y.X. Zheng & L.F. Liu	OR261306	OR261494	OR261682	OR261870
*C. aconidialis*	CSF16499	AACB	C	20190704-3-(20)	soil under 3-year-old *E. urophylla × E. grandis*	S.F. Chen, Q.C. Wang, W.X. Wu, Y.X. Zheng & L.F. Liu	OR261307	OR261495	OR261683	OR261871
*C. aconidialis*	CSF16502	AAAA	C	20190704-3-(25)	soil under 3-year-old *E. urophylla × E. grandis*	S.F. Chen, Q.C. Wang, W.X. Wu, Y.X. Zheng & L.F. Liu	OR261308	OR261496	OR261684	OR261872
*C. aconidialis*	CSF16507 ^b,c^	BAAA	C	20190704-3-(27)	soil under 3-year-old *E. urophylla × E. grandis*	S.F. Chen, Q.C. Wang, W.X. Wu, Y.X. Zheng & L.F. Liu	OR261309	OR261497	OR261685	OR261873
*C. aconidialis*	CSF16509	BAAA	C	20190704-3-(28)	soil under 3-year-old *E. urophylla × E. grandis*	S.F. Chen, Q.C. Wang, W.X. Wu, Y.X. Zheng & L.F. Liu	OR261310	OR261498	OR261686	OR261874
*C. aconidialis*	CSF16511	BAAA	C	20190704-3-(29)	soil under 3-year-old *E. urophylla × E. grandis*	S.F. Chen, Q.C. Wang, W.X. Wu, Y.X. Zheng & L.F. Liu	OR261311	OR261499	OR261687	OR261875
*C. aconidialis*	CSF16514	AAAA	C	20190704-3-(34)	soil under 3-year-old *E. urophylla × E. grandis*	S.F. Chen, Q.C. Wang, W.X. Wu, Y.X. Zheng & L.F. Liu	OR261312	OR261500	OR261688	OR261876
*C. aconidialis*	CSF16518	BAAA	C	20190704-3-(37)	soil under 3-year-old *E. urophylla × E. grandis*	S.F. Chen, Q.C. Wang, W.X. Wu, Y.X. Zheng & L.F. Liu	OR261313	OR261501	OR261689	OR261877
*C. aconidialis*	CSF16520 ^b,c^	ABAA	C	20190704-3-(38)	soil under 3-year-old *E. urophylla × E. grandis*	S.F. Chen, Q.C. Wang, W.X. Wu, Y.X. Zheng & L.F. Liu	OR261314	OR261502	OR261690	OR261878
*C. aconidialis*	CSF16522 ^b,c^	ADAA	C	20190704-3-(41)	soil under 3-year-old *E. urophylla × E. grandis*	S.F. Chen, Q.C. Wang, W.X. Wu, Y.X. Zheng & L.F. Liu	OR261315	OR261503	OR261691	OR261879
*C. aconidialis*	CSF16525	AHAA	C	20190704-3-(42)	soil under 3-year-old *E. urophylla × E. grandis*	S.F. Chen, Q.C. Wang, W.X. Wu, Y.X. Zheng & L.F. Liu	OR261316	OR261504	OR261692	OR261880
*C. aconidialis*	CSF16527 ^b,c^	AGBA	C	20190704-3-(43)	soil under 3-year-old *E. urophylla × E. grandis*	S.F. Chen, Q.C. Wang, W.X. Wu, Y.X. Zheng & L.F. Liu	OR261317	OR261505	OR261693	OR261881
*C. aconidialis*	CSF16530	AAAA	C	20190704-3-(45)	soil under 3-year-old *E. urophylla × E. grandis*	S.F. Chen, Q.C. Wang, W.X. Wu, Y.X. Zheng & L.F. Liu	OR261318	OR261506	OR261694	OR261882
*C. aconidialis*	CSF16533	BAAA	C	20190704-3-(48)	soil under 3-year-old *E. urophylla × E. grandis*	S.F. Chen, Q.C. Wang, W.X. Wu, Y.X. Zheng & L.F. Liu	OR261319	OR261507	OR261695	OR261883
*C. aconidialis*	CSF16535	BAAA	C	20190704-3-(49)	soil under 3-year-old *E. urophylla × E. grandis*	S.F. Chen, Q.C. Wang, W.X. Wu, Y.X. Zheng & L.F. Liu	OR261320	OR261508	OR261696	OR261884
*C. aconidialis*	CSF16537	AAAA	C	20190704-3-(51)	soil under 3-year-old *E. urophylla × E. grandis*	S.F. Chen, Q.C. Wang, W.X. Wu, Y.X. Zheng & L.F. Liu	OR261321	OR261509	OR261697	OR261885
*C. aconidialis*	CSF16539	AGBA	C	20190704-3-(52)	soil under 3-year-old *E. urophylla × E. grandis*	S.F. Chen, Q.C. Wang, W.X. Wu, Y.X. Zheng & L.F. Liu	OR261322	OR261510	OR261698	OR261886
*C. aconidialis*	CSF16540	AAAA	C	20190704-3-(53)	soil under 3-year-old *E. urophylla × E. grandis*	S.F. Chen, Q.C. Wang, W.X. Wu, Y.X. Zheng & L.F. Liu	OR261323	OR261511	OR261699	OR261887
*C. aconidialis*	CSF16542	AAAA	C	20190704-3-(54)	soil under 3-year-old *E. urophylla × E. grandis*	S.F. Chen, Q.C. Wang, W.X. Wu, Y.X. Zheng & L.F. Liu	OR261324	OR261512	OR261700	OR261888
*C. aconidialis*	CSF16544	AAAA	C	20190704-3-(55)	soil under 3-year-old *E. urophylla × E. grandis*	S.F. Chen, Q.C. Wang, W.X. Wu, Y.X. Zheng & L.F. Liu	OR261325	OR261513	OR261701	OR261889
*C. aconidialis*	CSF16546 ^b^	ABAA	C	20190704-3-(56)	soil under 3-year-old *E. urophylla × E. grandis*	S.F. Chen, Q.C. Wang, W.X. Wu, Y.X. Zheng & L.F. Liu	OR261326	OR261514	OR261702	OR261890
*C. aconidialis*	CSF16549	AHAA	C	20190704-3-(57)	soil under 3-year-old *E. urophylla × E. grandis*	S.F. Chen, Q.C. Wang, W.X. Wu, Y.X. Zheng & L.F. Liu	OR261327	OR261515	OR261703	OR261891
*C. aconidialis*	CSF16551	AHAA	C	20190704-3-(58)	soil under 3-year-old *E. urophylla × E. grandis*	S.F. Chen, Q.C. Wang, W.X. Wu, Y.X. Zheng & L.F. Liu	OR261328	OR261516	OR261704	OR261892
*C. aconidialis*	CSF16552	BAAA	C	20190704-3-(59)	soil under 3-year-old *E. urophylla × E. grandis*	S.F. Chen, Q.C. Wang, W.X. Wu, Y.X. Zheng & L.F. Liu	OR261329	OR261517	OR261705	OR261893
*C. aconidialis*	CSF16555	AAAA	C	20190704-3-(60)	soil under 3-year-old *E. urophylla × E. grandis*	S.F. Chen, Q.C. Wang, W.X. Wu, Y.X. Zheng & L.F. Liu	OR261330	OR261518	OR261706	OR261894
*C. aconidialis*	CSF16557 ^b,c^	CAAA	D	20190704-4-(1)	soil under 3-year-old *E. urophylla × E. grandis*	S.F. Chen, Q.C. Wang, W.X. Wu, Y.X. Zheng & L.F. Liu	OR261331	OR261519	OR261707	OR261895
*C. aconidialis*	CSF16561	AAAA	D	20190704-4-(2)	soil under 3-year-old *E. urophylla × E. grandis*	S.F. Chen, Q.C. Wang, W.X. Wu, Y.X. Zheng & L.F. Liu	OR261332	OR261520	OR261708	OR261896
*C. aconidialis*	CSF16562	BAAA	D	20190704-4-(4)	soil under 3-year-old *E. urophylla × E. grandis*	S.F. Chen, Q.C. Wang, W.X. Wu, Y.X. Zheng & L.F. Liu	OR261333	OR261521	OR261709	OR261897
*C. aconidialis*	CSF16564	BAAA	D	20190704-4-(5)	soil under 3-year-old *E. urophylla × E. grandis*	S.F. Chen, Q.C. Wang, W.X. Wu, Y.X. Zheng & L.F. Liu	OR261334	OR261522	OR261710	OR261898
*C. aconidialis*	CSF16566	AAAA	D	20190704-4-(7)	soil under 3-year-old *E. urophylla × E. grandis*	S.F. Chen, Q.C. Wang, W.X. Wu, Y.X. Zheng & L.F. Liu	OR261335	OR261523	OR261711	OR261899
*C. aconidialis*	CSF16568	AAAA	D	20190704-4-(8)	soil under 3-year-old *E. urophylla × E. grandis*	S.F. Chen, Q.C. Wang, W.X. Wu, Y.X. Zheng & L.F. Liu	OR261336	OR261524	OR261712	OR261900
*C. aconidialis*	CSF16571	AAAA	D	20190704-4-(9)	soil under 3-year-old *E. urophylla × E. grandis*	S.F. Chen, Q.C. Wang, W.X. Wu, Y.X. Zheng & L.F. Liu	OR261337	OR261525	OR261713	OR261901
*C. aconidialis*	CSF16573	AAAA	D	20190704-4-(10)	soil under 3-year-old *E. urophylla × E. grandis*	S.F. Chen, Q.C. Wang, W.X. Wu, Y.X. Zheng & L.F. Liu	OR261338	OR261526	OR261714	OR261902
*C. aconidialis*	CSF16575	AAAA	D	20190704-4-(11)	soil under 3-year-old *E. urophylla × E. grandis*	S.F. Chen, Q.C. Wang, W.X. Wu, Y.X. Zheng & L.F. Liu	OR261339	OR261527	OR261715	OR261903
*C. aconidialis*	CSF16578	AAAA	D	20190704-4-(12)	soil under 3-year-old *E. urophylla × E. grandis*	S.F. Chen, Q.C. Wang, W.X. Wu, Y.X. Zheng & L.F. Liu	OR261340	OR261528	OR261716	OR261904
*C. aconidialis*	CSF16580	AAAA	D	20190704-4-(15)	soil under 3-year-old *E. urophylla × E. grandis*	S.F. Chen, Q.C. Wang, W.X. Wu, Y.X. Zheng & L.F. Liu	OR261341	OR261529	OR261717	OR261905
*C. aconidialis*	CSF16582 ^b,c^	BEAA	D	20190704-4-(18)	soil under 3-year-old *E. urophylla × E. grandis*	S.F. Chen, Q.C. Wang, W.X. Wu, Y.X. Zheng & L.F. Liu	OR261342	OR261530	OR261718	OR261906
C. aconidialis	CSF16584 ^b,c^	BIAA	D	20190704-4-(22)	soil under 3-year-old *E. urophylla × E. grandis*	S.F. Chen, Q.C. Wang, W.X. Wu, Y.X. Zheng & L.F. Liu	OR261343	OR261531	OR261719	OR261907
*C. aconidialis*	CSF16586	AAAA	D	20190704-4-(23)	soil under 3-year-old *E. urophylla × E. grandis*	S.F. Chen, Q.C. Wang, W.X. Wu, Y.X. Zheng & L.F. Liu	OR261344	OR261532	OR261720	OR261908
*C. aconidialis*	CSF16588	BAAC	D	20190704-4-(24)	soil under 3-year-old *E. urophylla × E. grandis*	S.F. Chen, Q.C. Wang, W.X. Wu, Y.X. Zheng & L.F. Liu	OR261345	OR261533	OR261721	OR261909
*C. aconidialis*	CSF16591 ^b^	AGBA	D	20190704-4-(26)	soil under 3-year-old *E. urophylla × E. grandis*	S.F. Chen, Q.C. Wang, W.X. Wu, Y.X. Zheng & L.F. Liu	OR261346	OR261534	OR261722	OR261910
*C. aconidialis*	CSF16593	AAAA	D	20190704-4-(27)	soil under 3-year-old *E. urophylla × E. grandis*	S.F. Chen, Q.C. Wang, W.X. Wu, Y.X. Zheng & L.F. Liu	OR261347	OR261535	OR261723	OR261911
*C. aconidialis*	CSF16594	BAAA	D	20190704-4-(28)	soil under 3-year-old *E. urophylla × E. grandis*	S.F. Chen, Q.C. Wang, W.X. Wu, Y.X. Zheng & L.F. Liu	OR261348	OR261536	OR261724	OR261912
*C. aconidialis*	CSF16597	AAAA	D	20190704-4-(32)	soil under 3-year-old *E. urophylla × E. grandis*	S.F. Chen, Q.C. Wang, W.X. Wu, Y.X. Zheng & L.F. Liu	OR261349	OR261537	OR261725	OR261913
*C. aconidialis*	CSF16599 ^b,c^	AACB	D	20190704-4-(34)	soil under 3-year-old *E. urophylla × E. grandis*	S.F. Chen, Q.C. Wang, W.X. Wu, Y.X. Zheng & L.F. Liu	OR261350	OR261538	OR261726	OR261914
*C. aconidialis*	CSF16602	AACB	D	20190704-4-(36)	soil under 3-year-old *E. urophylla × E. grandis*	S.F. Chen, Q.C. Wang, W.X. Wu, Y.X. Zheng & L.F. Liu	OR261351	OR261539	OR261727	OR261915
*C. aconidialis*	CSF16604	BAAA	D	20190704-4-(37)	soil under 3-year-old *E. urophylla × E. grandis*	S.F. Chen, Q.C. Wang, W.X. Wu, Y.X. Zheng & L.F. Liu	OR261352	OR261540	OR261728	OR261916
*C. aconidialis*	CSF16607 ^b^	AACB	D	20190704-4-(38)	soil under 3-year-old *E. urophylla × E. grandis*	S.F. Chen, Q.C. Wang, W.X. Wu, Y.X. Zheng & L.F. Liu	OR261353	OR261541	OR261729	OR261917
*C. aconidialis*	CSF16609 ^b,c^	AEAA	D	20190704-4-(40)	soil under 3-year-old *E. urophylla × E. grandis*	S.F. Chen, Q.C. Wang, W.X. Wu, Y.X. Zheng & L.F. Liu	OR261354	OR261542	OR261730	OR261918
*C. aconidialis*	CSF16612	BAAA	D	20190704-4-(44)	soil under 3-year-old *E. urophylla × E. grandis*	S.F. Chen, Q.C. Wang, W.X. Wu, Y.X. Zheng & L.F. Liu	OR261355	OR261543	OR261731	OR261919
*C. aconidialis*	CSF16614	AAAA	D	20190704-4-(45)	soil under 3-year-old *E. urophylla × E. grandis*	S.F. Chen, Q.C. Wang, W.X. Wu, Y.X. Zheng & L.F. Liu	OR261356	OR261544	OR261732	OR261920
*C. aconidialis*	CSF16618	AACB	D	20190704-4-(46)	soil under 3-year-old *E. urophylla × E. grandis*	S.F. Chen, Q.C. Wang, W.X. Wu, Y.X. Zheng & L.F. Liu	OR261357	OR261545	OR261733	OR261921
*C. aconidialis*	CSF16621	BAAA	D	20190704-4-(47)	soil under 3-year-old *E. urophylla × E. grandis*	S.F. Chen, Q.C. Wang, W.X. Wu, Y.X. Zheng & L.F. Liu	OR261358	OR261546	OR261734	OR261922
*C. aconidialis*	CSF16625	BAAA	D	20190704-4-(48)	soil under 3-year-old *E. urophylla × E. grandis*	S.F. Chen, Q.C. Wang, W.X. Wu, Y.X. Zheng & L.F. Liu	OR261359	OR261547	OR261735	OR261923
*C. aconidialis*	CSF16627 ^b^	AFBA	D	20190704-4-(50)	soil under 3-year-old *E. urophylla × E. grandis*	S.F. Chen, Q.C. Wang, W.X. Wu, Y.X. Zheng & L.F. Liu	OR261360	OR261548	OR261736	OR261924
*C. aconidialis*	CSF16631	BAAA	D	20190704-4-(51)	soil under 3-year-old *E. urophylla × E. grandis*	S.F. Chen, Q.C. Wang, W.X. Wu, Y.X. Zheng & L.F. Liu	OR261361	OR261549	OR261737	OR261925
*C. aconidialis*	CSF16633	AAAA	D	20190704-4-(52)	soil under 3-year-old *E. urophylla × E. grandis*	S.F. Chen, Q.C. Wang, W.X. Wu, Y.X. Zheng & L.F. Liu	OR261362	OR261550	OR261738	OR261926
*C. aconidialis*	CSF16640	BAAC	E	20190705-1-(4)	soil under 3-year-old *E. urophylla × E. grandis*	S.F. Chen, Q.C. Wang, W.X. Wu, Y.X. Zheng & L.F. Liu	OR261363	OR261551	OR261739	OR261927
*C. aconidialis*	CSF16643 ^b,c^	AGBD	E	20190705-1-(10)	soil under 3-year-old *E. urophylla × E. grandis*	S.F. Chen, Q.C. Wang, W.X. Wu, Y.X. Zheng & L.F. Liu	OR261364	OR261552	OR261740	OR261928
*C. aconidialis*	CSF16645	AAAA	E	20190705-1-(11)	soil under 3-year-old *E. urophylla × E. grandis*	S.F. Chen, Q.C. Wang, W.X. Wu, Y.X. Zheng & L.F. Liu	OR261365	OR261553	OR261741	OR261929
*C. aconidialis*	CSF16648 ^b,c^	BACA	E	20190705-1-(12)	soil under 3-year-old *E. urophylla × E. grandis*	S.F. Chen, Q.C. Wang, W.X. Wu, Y.X. Zheng & L.F. Liu	OR261366	OR261554	OR261742	OR261930
*C. aconidialis*	CSF16651	BAAC	E	20190705-1-(13)	soil under 3-year-old *E. urophylla × E. grandis*	S.F. Chen, Q.C. Wang, W.X. Wu, Y.X. Zheng & L.F. Liu	OR261367	OR261555	OR261743	OR261931
*C. aconidialis*	CSF16653 ^b,c^	BACB	E	20190705-1-(14)	soil under 3-year-old *E. urophylla × E. grandis*	S.F. Chen, Q.C. Wang, W.X. Wu, Y.X. Zheng & L.F. Liu	OR261368	OR261556	OR261744	OR261932
*C. aconidialis*	CSF16655	AAAA	E	20190705-1-(15)	soil under 3-year-old *E. urophylla × E. grandis*	S.F. Chen, Q.C. Wang, W.X. Wu, Y.X. Zheng & L.F. Liu	OR261369	OR261557	OR261745	OR261933
*C. aconidialis*	CSF16657	BAAB	E	20190705-1-(17)	soil under 3-year-old *E. urophylla × E. grandis*	S.F. Chen, Q.C. Wang, W.X. Wu, Y.X. Zheng & L.F. Liu	OR261370	OR261558	OR261746	OR261934
*C. aconidialis*	CSF16659	BAAA	E	20190705-1-(22)	soil under 3-year-old *E. urophylla × E. grandis*	S.F. Chen, Q.C. Wang, W.X. Wu, Y.X. Zheng & L.F. Liu	OR261371	OR261559	OR261747	OR261935
*C. aconidialis*	CSF16661	AAAA	E	20190705-1-(23)	soil under 3-year-old *E. urophylla × E. grandis*	S.F. Chen, Q.C. Wang, W.X. Wu, Y.X. Zheng & L.F. Liu	OR261372	OR261560	OR261748	OR261936
*C. aconidialis*	CSF16663	AAAA	E	20190705-1-(27)	soil under 3-year-old *E. urophylla × E. grandis*	S.F. Chen, Q.C. Wang, W.X. Wu, Y.X. Zheng & L.F. Liu	OR261373	OR261561	OR261749	OR261937
*C. aconidialis*	CSF16666 ^b^	AGBD	E	20190705-1-(28)	soil under 3-year-old *E. urophylla × E. grandis*	S.F. Chen, Q.C. Wang, W.X. Wu, Y.X. Zheng & L.F. Liu	OR261374	OR261562	OR261750	OR261938
*C. aconidialis*	CSF16668	AAAA	E	20190705-1-(31)	soil under 3-year-old *E. urophylla × E. grandis*	S.F. Chen, Q.C. Wang, W.X. Wu, Y.X. Zheng & L.F. Liu	OR261375	OR261563	OR261751	OR261939
*C. aconidialis*	CSF16672	AFBA	E	20190705-1-(38)	soil under 3-year-old *E. urophylla × E. grandis*	S.F. Chen, Q.C. Wang, W.X. Wu, Y.X. Zheng & L.F. Liu	OR261376	OR261564	OR261752	OR261940
*C. aconidialis*	CSF16675 ^b,c^	AACA	E	20190705-1-(39)	soil under 3-year-old *E. urophylla × E. grandis*	S.F. Chen, Q.C. Wang, W.X. Wu, Y.X. Zheng & L.F. Liu	OR261377	OR261565	OR261753	OR261941
*C. aconidialis*	CSF16677	BAAB	E	20190705-1-(40)	soil under 3-year-old *E. urophylla × E. grandis*	S.F. Chen, Q.C. Wang, W.X. Wu, Y.X. Zheng & L.F. Liu	OR261378	OR261566	OR261754	OR261942
*C. aconidialis*	CSF16682	AAAA	E	20190705-1-(43)	soil under 3-year-old *E. urophylla × E. grandis*	S.F. Chen, Q.C. Wang, W.X. Wu, Y.X. Zheng & L.F. Liu	OR261379	OR261567	OR261755	OR261943
*C. aconidialis*	CSF16686	AAAA	E	20190705-1-(45)	soil under 3-year-old *E. urophylla × E. grandis*	S.F. Chen, Q.C. Wang, W.X. Wu, Y.X. Zheng & L.F. Liu	OR261380	OR261568	OR261756	OR261944
*C. aconidialis*	CSF16689	BAAB	E	20190705-1-(46)	soil under 3-year-old *E. urophylla × E. grandis*	S.F. Chen, Q.C. Wang, W.X. Wu, Y.X. Zheng & L.F. Liu	OR261381	OR261569	OR261757	OR261945
*C. aconidialis*	CSF16691	AAAA	E	20190705-1-(48)	soil under 3-year-old *E. urophylla × E. grandis*	S.F. Chen, Q.C. Wang, W.X. Wu, Y.X. Zheng & L.F. Liu	OR261382	OR261570	OR261758	OR261946
*C. aconidialis*	CSF16693 ^b,c^	BAAB	E	20190705-1-(49)	soil under 3-year-old *E. urophylla × E. grandis*	S.F. Chen, Q.C. Wang, W.X. Wu, Y.X. Zheng & L.F. Liu	OR261383	OR261571	OR261759	OR261947
*C. aconidialis*	CSF16695	BAAB	E	20190705-1-(50)	soil under 3-year-old *E. urophylla × E. grandis*	S.F. Chen, Q.C. Wang, W.X. Wu, Y.X. Zheng & L.F. Liu	OR261384	OR261572	OR261760	OR261948
*C. aconidialis*	CSF16697	BAAA	E	20190705-1-(52)	soil under 3-year-old *E. urophylla × E. grandis*	S.F. Chen, Q.C. Wang, W.X. Wu, Y.X. Zheng & L.F. Liu	OR261385	OR261573	OR261761	OR261949
*C. aconidialis*	CSF16702	BAAA	E	20190705-1-(54)	soil under 3-year-old *E. urophylla × E. grandis*	S.F. Chen, Q.C. Wang, W.X. Wu, Y.X. Zheng & L.F. Liu	OR261386	OR261574	OR261762	OR261950
*C. aconidialis*	CSF16704	BAAA	E	20190705-1-(55)	soil under 3-year-old *E. urophylla × E. grandis*	S.F. Chen, Q.C. Wang, W.X. Wu, Y.X. Zheng & L.F. Liu	OR261387	OR261575	OR261763	OR261951
*C. aconidialis*	CSF16706 ^b,c^	BAAC	E	20190705-1-(56)	soil under 3-year-old *E. urophylla × E. grandis*	S.F. Chen, Q.C. Wang, W.X. Wu, Y.X. Zheng & L.F. Liu	OR261388	OR261576	OR261764	OR261952
*C. aconidialis*	CSF16707	AAAA	E	20190705-1-(58)	soil under 3-year-old *E. urophylla × E. grandis*	S.F. Chen, Q.C. Wang, W.X. Wu, Y.X. Zheng & L.F. Liu	OR261389	OR261577	OR261765	OR261953
*C. aconidialis*	CSF16710	AAAA	E	20190705-1-(59)	soil under 3-year-old *E. urophylla × E. grandis*	S.F. Chen, Q.C. Wang, W.X. Wu, Y.X. Zheng & L.F. Liu	OR261390	OR261578	OR261766	OR261954
*C. aconidialis*	CSF16712	AAAA	E	20190705-1-(60)	soil under 3-year-old *E. urophylla × E. grandis*	S.F. Chen, Q.C. Wang, W.X. Wu, Y.X. Zheng & L.F. Liu	OR261391	OR261579	OR261767	OR261955
*C. aconidialis*	CSF16714	AAAA	E	20190705-1-(61)	soil under 3-year-old *E. urophylla × E. grandis*	S.F. Chen, Q.C. Wang, W.X. Wu, Y.X. Zheng & L.F. Liu	OR261392	OR261580	OR261768	OR261956
*C. aconidialis*	CSF16716	AAAA	E	20190705-1-(62)	soil under 3-year-old *E. urophylla × E. grandis*	S.F. Chen, Q.C. Wang, W.X. Wu, Y.X. Zheng & L.F. Liu	OR261393	OR261581	OR261769	OR261957
*C. aconidialis*	CSF16718 ^b,c^	ACAA	E	20190705-1-(64)	soil under 3-year-old *E. urophylla × E. grandis*	S.F. Chen, Q.C. Wang, W.X. Wu, Y.X. Zheng & L.F. Liu	OR261394	OR261582	OR261770	OR261958
*C. aconidialis*	CSF16720	BAAC	E	20190705-1-(67)	soil under 3-year-old *E. urophylla × E. grandis*	S.F. Chen, Q.C. Wang, W.X. Wu, Y.X. Zheng & L.F. Liu	OR261395	OR261583	OR261771	OR261959
*C. aconidialis*	CSF16722	BAAA	E	20190705-1-(69)	soil under 3-year-old *E. urophylla × E. grandis*	S.F. Chen, Q.C. Wang, W.X. Wu, Y.X. Zheng & L.F. Liu	OR261396	OR261584	OR261772	OR261960
*C. aconidialis*	CSF16728	AAAA	F	20190705-2-(5)	soil under 5-year-old *E. urophylla × E. grandis*	S.F. Chen, Q.C. Wang, W.X. Wu, Y.X. Zheng & L.F. Liu	OR261397	OR261585	OR261773	OR261961
*C. aconidialis*	CSF16729	BAAA	F	20190705-2-(8)	soil under 5-year-old *E. urophylla × E. grandis*	S.F. Chen, Q.C. Wang, W.X. Wu, Y.X. Zheng & L.F. Liu	OR261398	OR261586	OR261774	OR261962
*C. aconidialis*	CSF16735 ^b^	BAAA	F	20190705-2-(14)	soil under 5-year-old *E. urophylla × E. grandis*	S.F. Chen, Q.C. Wang, W.X. Wu, Y.X. Zheng & L.F. Liu	OR261399	OR261587	OR261775	OR261963
*C. aconidialis*	CSF16739	AAAA	F	20190705-2-(17)	soil under 5-year-old *E. urophylla × E. grandis*	S.F. Chen, Q.C. Wang, W.X. Wu, Y.X. Zheng & L.F. Liu	OR261400	OR261588	OR261776	OR261964
*C. aconidialis*	CSF16742 ^b,c^	AFBA	F	20190705-2-(18)	soil under 5-year-old *E. urophylla × E. grandis*	S.F. Chen, Q.C. Wang, W.X. Wu, Y.X. Zheng & L.F. Liu	OR261401	OR261589	OR261777	OR261965
*C. aconidialis*	CSF16751	AGBA	F	20190705-2-(21)	soil under 5-year-old *E. urophylla × E. grandis*	S.F. Chen, Q.C. Wang, W.X. Wu, Y.X. Zheng & L.F. Liu	OR261402	OR261590	OR261778	OR261966
*C. aconidialis*	CSF16760	BAAA	F	20190705-2-(26)	soil under 5-year-old *E. urophylla × E. grandis*	S.F. Chen, Q.C. Wang, W.X. Wu, Y.X. Zheng & L.F. Liu	OR261403	OR261591	OR261779	OR261967
*C. aconidialis*	CSF16762	AFBA	F	20190705-2-(27)	soil under 5-year-old *E. urophylla × E. grandis*	S.F. Chen, Q.C. Wang, W.X. Wu, Y.X. Zheng & L.F. Liu	OR261404	OR261592	OR261780	OR261968
*C. aconidialis*	CSF16767	BAAA	F	20190705-2-(29)	soil under 5-year-old *E. urophylla × E. grandis*	S.F. Chen, Q.C. Wang, W.X. Wu, Y.X. Zheng & L.F. Liu	OR261405	OR261593	OR261781	OR261969
*C. aconidialis*	CSF16770	AFBA	F	20190705-2-(30)	soil under 5-year-old *E. urophylla × E. grandis*	S.F. Chen, Q.C. Wang, W.X. Wu, Y.X. Zheng & L.F. Liu	OR261406	OR261594	OR261782	OR261970
*C. aconidialis*	CSF16774	AAAA	F	20190705-2-(31)	soil under 5-year-old *E. urophylla × E. grandis*	S.F. Chen, Q.C. Wang, W.X. Wu, Y.X. Zheng & L.F. Liu	OR261407	OR261595	OR261783	OR261971
*C. aconidialis*	CSF16779	AAAA	F	20190705-2-(33)	soil under 5-year-old *E. urophylla × E. grandis*	S.F. Chen, Q.C. Wang, W.X. Wu, Y.X. Zheng & L.F. Liu	OR261408	OR261596	OR261784	OR261972
*C. aconidialis*	CSF16788	AAAA	F	20190705-2-(38)	soil under 5-year-old *E. urophylla × E. grandis*	S.F. Chen, Q.C. Wang, W.X. Wu, Y.X. Zheng & L.F. Liu	OR261409	OR261597	OR261785	OR261973
*C. aconidialis*	CSF16792 ^b^	BAAC	F	20190705-2-(41)	soil under 5-year-old *E. urophylla × E. grandis*	S.F. Chen, Q.C. Wang, W.X. Wu, Y.X. Zheng & L.F. Liu	OR261410	OR261598	OR261786	OR261974
*C. aconidialis*	CSF16809 ^b^	BAAB	F	20190705-2-(55)	soil under 5-year-old *E. urophylla × E. grandis*	S.F. Chen, Q.C. Wang, W.X. Wu, Y.X. Zheng & L.F. Liu	OR261411	OR261599	OR261787	OR261975
*C. aconidialis*	CSF17104	AHAA	A	20190806-2-(1)	soil under 4-year-old *E. urophylla × E. grandis*	S.F. Chen, Q.C. Wang, L.L. Liu, Y. Liu, Y.C. Qu, Y.L. Li & X.Y. Liang	OR261412	OR261600	OR261788	OR261976
*C. aconidialis*	CSF17110 ^b,c^	AAAA	A	20190806-2-(5)	soil under 4-year-old *E. urophylla × E. grandis*	S.F. Chen, Q.C. Wang, L.L. Liu, Y. Liu, Y.C. Qu, Y.L. Li & X.Y. Liang	OR261413	OR261601	OR261789	OR261977
*C. aconidialis*	CSF17112	BAAA	A	20190806-2-(7)	soil under 4-year-old *E. urophylla × E. grandis*	S.F. Chen, Q.C. Wang, L.L. Liu, Y. Liu, Y.C. Qu, Y.L. Li & X.Y. Liang	OR261414	OR261602	OR261790	OR261978
*C. aconidialis*	CSF17114	AAAA	A	20190806-2-(8)	soil under 4-year-old *E. urophylla × E. grandis*	S.F. Chen, Q.C. Wang, L.L. Liu, Y. Liu, Y.C. Qu, Y.L. Li & X.Y. Liang	OR261415	OR261603	OR261791	OR261979
*C. aconidialis*	CSF17116	AHAA	A	20190806-2-(9)	soil under 4-year-old *E. urophylla × E. grandis*	S.F. Chen, Q.C. Wang, L.L. Liu, Y. Liu, Y.C. Qu, Y.L. Li & X.Y. Liang	OR261416	OR261604	OR261792	OR261980
*C. aconidialis*	CSF17130 ^c^	AAAA	A	20190806-2-(24)	soil under 4-year-old *E. urophylla × E. grandis*	S.F. Chen, Q.C. Wang, L.L. Liu, Y. Liu, Y.C. Qu, Y.L. Li & X.Y. Liang	OR261417	OR261605	OR261793	OR261981
*C. aconidialis*	CSF17133	AAAA	A	20190806-2-(25)	soil under 4-year-old *E. urophylla × E. grandis*	S.F. Chen, Q.C. Wang, L.L. Liu, Y. Liu, Y.C. Qu, Y.L. Li & X.Y. Liang	OR261418	OR261606	OR261794	OR261982
*C. aconidialis*	CSF17135	AHAA	A	20190806-2-(27)	soil under 4-year-old *E. urophylla × E. grandis*	S.F. Chen, Q.C. Wang, L.L. Liu, Y. Liu, Y.C. Qu, Y.L. Li & X.Y. Liang	OR261419	OR261607	OR261795	OR261983
*C. aconidialis*	CSF17137	AAAA	A	20190806-2-(28)	soil under 4-year-old *E. urophylla × E. grandis*	S.F. Chen, Q.C. Wang, L.L. Liu, Y. Liu, Y.C. Qu, Y.L. Li & X.Y. Liang	OR261420	OR261608	OR261796	OR261984
*C. aconidialis*	CSF17140	AAAA	A	20190806-2-(31)	soil under 4-year-old *E. urophylla × E. grandis*	S.F. Chen, Q.C. Wang, L.L. Liu, Y. Liu, Y.C. Qu, Y.L. Li & X.Y. Liang	OR261421	OR261609	OR261797	OR261985
*C. aconidialis*	CSF17142 ^b,c^	AAAA	A	20190806-2-(38)	soil under 4-year-old *E. urophylla × E. grandis*	S.F. Chen, Q.C. Wang, L.L. Liu, Y. Liu, Y.C. Qu, Y.L. Li & X.Y. Liang	OR261422	OR261610	OR261798	OR261986
*C. aconidialis*	CSF17144	AAAA	A	20190806-2-(41)	soil under 4-year-old *E. urophylla × E. grandis*	S.F. Chen, Q.C. Wang, L.L. Liu, Y. Liu, Y.C. Qu, Y.L. Li & X.Y. Liang	OR261423	OR261611	OR261799	OR261987
*C. aconidialis*	CSF17146	AAAA	A	20190806-2-(42)	soil under 4-year-old *E. urophylla × E. grandis*	S.F. Chen, Q.C. Wang, L.L. Liu, Y. Liu, Y.C. Qu, Y.L. Li & X.Y. Liang	OR261424	OR261612	OR261800	OR261988
*C. aconidialis*	CSF17150	AAAA	A	20190806-2-(44)	soil under 4-year-old *E. urophylla × E. grandis*	S.F. Chen, Q.C. Wang, L.L. Liu, Y. Liu, Y.C. Qu, Y.L. Li & X.Y. Liang	OR261425	OR261613	OR261801	OR261989
*C. aconidialis*	CSF17153	AAAA	A	20190806-2-(45)	soil under 4-year-old *E. urophylla × E. grandis*	S.F. Chen, Q.C. Wang, L.L. Liu, Y. Liu, Y.C. Qu, Y.L. Li & X.Y. Liang	OR261426	OR261614	OR261802	OR261990
*C. aconidialis*	CSF17155	AAAA	A	20190806-2-(46)	soil under 4-year-old *E. urophylla × E. grandis*	S.F. Chen, Q.C. Wang, L.L. Liu, Y. Liu, Y.C. Qu, Y.L. Li & X.Y. Liang	OR261427	OR261615	OR261803	OR261991
*C. aconidialis*	CSF17158	AAAA	A	20190806-2-(47)	soil under 4-year-old *E. urophylla × E. grandis*	S.F. Chen, Q.C. Wang, L.L. Liu, Y. Liu, Y.C. Qu, Y.L. Li & X.Y. Liang	OR261428	OR261616	OR261804	OR261992
*C. aconidialis*	CSF17160	AAAA	A	20190806-2-(49)	soil under 4-year-old *E. urophylla × E. grandis*	S.F. Chen, Q.C. Wang, L.L. Liu, Y. Liu, Y.C. Qu, Y.L. Li & X.Y. Liang	OR261429	OR261617	OR261805	OR261993
*C. aconidialis*	CSF17163 ^b^	AHAA	A	20190806-2-(51)	soil under 4-year-old *E. urophylla × E. grandis*	S.F. Chen, Q.C. Wang, L.L. Liu, Y. Liu, Y.C. Qu, Y.L. Li & X.Y. Liang	OR261430	OR261618	OR261806	OR261994
*C. aconidialis*	CSF17166	BAAA	A	20190806-2-(52)	soil under 4-year-old *E. urophylla × E. grandis*	S.F. Chen, Q.C. Wang, L.L. Liu, Y. Liu, Y.C. Qu, Y.L. Li & X.Y. Liang	OR261431	OR261619	OR261807	OR261995
*C. aconidialis*	CSF17169	AAAA	A	20190806-2-(53)	soil under 4-year-old *E. urophylla × E. grandis*	S.F. Chen, Q.C. Wang, L.L. Liu, Y. Liu, Y.C. Qu, Y.L. Li & X.Y. Liang	OR261432	OR261620	OR261808	OR261996
*C. aconidialis*	CSF17172	AAAA	A	20190806-2-(54)	soil under 4-year-old *E. urophylla × E. grandis*	S.F. Chen, Q.C. Wang, L.L. Liu, Y. Liu, Y.C. Qu, Y.L. Li & X.Y. Liang	OR261433	OR261621	OR261809	OR261997
*C. aconidialis*	CSF17181	AAAA	A	20190806-2-(59)	soil under 4-year-old *E. urophylla × E. grandis*	S.F. Chen, Q.C. Wang, L.L. Liu, Y. Liu, Y.C. Qu, Y.L. Li & X.Y. Liang	OR261434	OR261622	OR261810	OR261998
*C. aconidialis*	CSF17184	AAAA	A	20190806-2-(60)	soil under 4-year-old *E. urophylla × E. grandis*	S.F. Chen, Q.C. Wang, L.L. Liu, Y. Liu, Y.C. Qu, Y.L. Li & X.Y. Liang	OR261435	OR261623	OR261811	OR261999
*C. aconidialis*	CSF17187	AAAA	A	20190806-2-(61)	soil under 4-year-old *E. urophylla × E. grandis*	S.F. Chen, Q.C. Wang, L.L. Liu, Y. Liu, Y.C. Qu, Y.L. Li & X.Y. Liang	OR261436	OR261624	OR261812	OR262000
*C. hongkongensis*	CSF16463 ^b,c^	AGAA	B	20190704-2-(6)	soil under 3-year-old *E. urophylla × E. grandis*	S.F. Chen, Q.C. Wang, W.X. Wu, Y.X. Zheng & L.F. Liu	OR261437	OR261625	OR261813	OR262001
*C. hongkongensis*	CSF16464	AAAA	B	20190704-2-(14)	soil under 3-year-old *E. urophylla × E. grandis*	S.F. Chen, Q.C. Wang, W.X. Wu, Y.X. Zheng & L.F. Liu	OR261438	OR261626	OR261814	OR262002
*C. hongkongensis*	CSF16486	AAAA	C	20190704-3-(12)	soil under 3-year-old *E. urophylla × E. grandis*	S.F. Chen, Q.C. Wang, W.X. Wu, Y.X. Zheng & L.F. Liu	OR261439	OR261627	OR261815	OR262003
*C. hongkongensis*	CSF16611	AAAA	D	20190704-4-(41)	soil under 3-year-old *E. urophylla × E. grandis*	S.F. Chen, Q.C. Wang, W.X. Wu, Y.X. Zheng & L.F. Liu	OR261440	OR261628	OR261816	OR262004
*C. hongkongensis*	CSF16637	AAAA	E	20190705-1-(3)	soil under 3-year-old *E. urophylla × E. grandis*	S.F. Chen, Q.C. Wang, W.X. Wu, Y.X. Zheng & L.F. Liu	OR261441	OR261629	OR261817	OR262005
*C. hongkongensis*	CSF16670	AAAA	E	20190705-1-(36)	soil under 3-year-old *E. urophylla × E. grandis*	S.F. Chen, Q.C. Wang, W.X. Wu, Y.X. Zheng & L.F. Liu	OR261442	OR261630	OR261818	OR262006
*C. hongkongensis*	CSF16680	AAAA	E	20190705-1-(42)	soil under 3-year-old *E. urophylla × E. grandis*	S.F. Chen, Q.C. Wang, W.X. Wu, Y.X. Zheng & L.F. Liu	OR261443	OR261631	OR261819	OR262007
*C. hongkongensis*	CSF16699	AAAA	E	20190705-1-(53)	soil under 3-year-old *E. urophylla × E. grandis*	S.F. Chen, Q.C. Wang, W.X. Wu, Y.X. Zheng & L.F. Liu	OR261444	OR261632	OR261820	OR262008
*C. hongkongensis*	CSF16726 ^b,c^	AAAA	F	20190705-2-(3)	soil under 5-year-old *E. urophylla × E. grandis*	S.F. Chen, Q.C. Wang, W.X. Wu, Y.X. Zheng & L.F. Liu	OR261445	OR261633	OR261821	OR262009
*C. hongkongensis*	CSF16731 ^b,c^	AABA	F	20190705-2-(11)	soil under 5-year-old *E. urophylla × E. grandis*	S.F. Chen, Q.C. Wang, W.X. Wu, Y.X. Zheng & L.F. Liu	OR261446	OR261634	OR261822	OR262010
*C. hongkongensis*	CSF16733	AAAA	F	20190705-2-(13)	soil under 5-year-old *E. urophylla × E. grandis*	S.F. Chen, Q.C. Wang, W.X. Wu, Y.X. Zheng & L.F. Liu	OR261447	OR261635	OR261823	OR262011
*C. hongkongensis*	CSF16737 ^b,c^	ABAA	F	20190705-2-(16)	soil under 5-year-old *E. urophylla × E. grandis*	S.F. Chen, Q.C. Wang, W.X. Wu, Y.X. Zheng & L.F. Liu	OR261448	OR261636	OR261824	OR262012
*C. hongkongensis*	CSF16745	AAAA	F	20190705-2-(19)	soil under 5-year-old *E. urophylla × E. grandis*	S.F. Chen, Q.C. Wang, W.X. Wu, Y.X. Zheng & L.F. Liu	OR261449	OR261637	OR261825	OR262013
*C. hongkongensis*	CSF16748	AAAA	F	20190705-2-(20)	soil under 5-year-old *E. urophylla × E. grandis*	S.F. Chen, Q.C. Wang, W.X. Wu, Y.X. Zheng & L.F. Liu	OR261450	OR261638	OR261826	OR262014
*C. hongkongensis*	CSF16754 ^b,c^	BAAA	F	20190705-2-(22)	soil under 5-year-old *E. urophylla × E. grandis*	S.F. Chen, Q.C. Wang, W.X. Wu, Y.X. Zheng & L.F. Liu	OR261451	OR261639	OR261827	OR262015
*C. hongkongensis*	CSF16756 ^c^	AAAA	F	20190705-2-(25)	soil under 5-year-old *E. urophylla × E. grandis*	S.F. Chen, Q.C. Wang, W.X. Wu, Y.X. Zheng & L.F. Liu	OR261452	OR261640	OR261828	OR262016
*C. hongkongensis*	CSF16765 ^b,c^	AFAA	F	20190705-2-(28)	soil under 5-year-old *E. urophylla × E. grandis*	S.F. Chen, Q.C. Wang, W.X. Wu, Y.X. Zheng & L.F. Liu	OR261453	OR261641	OR261829	OR262017
*C. hongkongensis*	CSF16781 ^b,c^	ADAA	F	20190705-2-(35)	soil under 5-year-old *E. urophylla × E. grandis*	S.F. Chen, Q.C. Wang, W.X. Wu, Y.X. Zheng & L.F. Liu	OR261454	OR261642	OR261830	OR262018
*C. hongkongensis*	CSF16786 ^b,c^	AEAA	F	20190705-2-(36)	soil under 5-year-old *E. urophylla × E. grandis*	S.F. Chen, Q.C. Wang, W.X. Wu, Y.X. Zheng & L.F. Liu	OR261455	OR261643	OR261831	OR262019
*C. hongkongensis*	CSF16790	AAAA	F	20190705-2-(39)	soil under 5-year-old *E. urophylla × E. grandis*	S.F. Chen, Q.C. Wang, W.X. Wu, Y.X. Zheng & L.F. Liu	OR261456	OR261644	OR261832	OR262020
*C. hongkongensis*	CSF16795	AAAA	F	20190705-2-(44)	soil under 5-year-old *E. urophylla × E. grandis*	S.F. Chen, Q.C. Wang, W.X. Wu, Y.X. Zheng & L.F. Liu	OR261457	OR261645	OR261833	OR262021
*C. hongkongensis*	CSF16797	AAAA	F	20190705-2-(46)	soil under 5-year-old *E. urophylla × E. grandis*	S.F. Chen, Q.C. Wang, W.X. Wu, Y.X. Zheng & L.F. Liu	OR261458	OR261646	OR261834	OR262022
*C. hongkongensis*	CSF16803	AAAA	F	20190705-2-(49)	soil under 5-year-old *E. urophylla × E. grandis*	S.F. Chen, Q.C. Wang, W.X. Wu, Y.X. Zheng & L.F. Liu	OR261459	OR261647	OR261835	OR262023
*C. hongkongensis*	CSF16805	AAAA	F	20190705-2-(52)	soil under 5-year-old *E. urophylla × E. grandis*	S.F. Chen, Q.C. Wang, W.X. Wu, Y.X. Zheng & L.F. Liu	OR261460	OR261648	OR261836	OR262024
*C. hongkongensis*	CSF16811	AAAA	F	20190705-2-(56)	soil under 5-year-old *E. urophylla × E. grandis*	S.F. Chen, Q.C. Wang, W.X. Wu, Y.X. Zheng & L.F. Liu	OR261461	OR261649	OR261837	OR262025
*C. hongkongensis*	CSF16813 ^b^	ABAA	F	20190705-2-(57)	soil under 5-year-old *E. urophylla × E. grandis*	S.F. Chen, Q.C. Wang, W.X. Wu, Y.X. Zheng & L.F. Liu	OR261462	OR261650	OR261838	OR262026
*C. hongkongensis*	CSF16816	AAAA	F	20190705-2-(58)	soil under 5-year-old *E. urophylla × E. grandis*	S.F. Chen, Q.C. Wang, W.X. Wu, Y.X. Zheng & L.F. Liu	OR261463	OR261651	OR261839	OR262027
*C. hongkongensis*	CSF16819	AAAA	F	20190705-2-(62)	soil under 5-year-old *E. urophylla × E. grandis*	S.F. Chen, Q.C. Wang, W.X. Wu, Y.X. Zheng & L.F. Liu	OR261464	OR261652	OR261840	OR262028
*C. hongkongensis*	CSF16821	AAAA	G	20190705-4-(5)	soil under 3-year-old *E. urophylla × E. grandis*	S.F. Chen, Q.C. Wang, W.X. Wu, Y.X. Zheng & L.F. Liu	OR261465	OR261653	OR261841	OR262029
*C. hongkongensis*	CSF16823 ^b,c^	AAAB	G	20190705-4-(14)	soil under 3-year-old *E. urophylla × E. grandis*	S.F. Chen, Q.C. Wang, W.X. Wu, Y.X. Zheng & L.F. Liu	OR261466	OR261654	OR261842	OR262030
*C. hongkongensis*	CSF17107	AAAA	A	20190806-2-(4)	soil under 4-year-old *E. urophylla × E. grandis*	S.F. Chen, Q.C. Wang, L.L. Liu, Y. Liu, Y.C. Qu, Y.L. Li & X.Y. Liang	OR261467	OR261655	OR261843	OR262031
*C. hongkongensis*	CSF17118 ^b,c^	ACAA	A	20190806-2-(11)	soil under 4-year-old *E. urophylla × E. grandis*	S.F. Chen, Q.C. Wang, L.L. Liu, Y. Liu, Y.C. Qu, Y.L. Li & X.Y. Liang	OR261468	OR261656	OR261844	OR262032
*C. hongkongensis*	CSF17120	AAAA	A	20190806-2-(13)	soil under 4-year-old *E. urophylla × E. grandis*	S.F. Chen, Q.C. Wang, L.L. Liu, Y. Liu, Y.C. Qu, Y.L. Li & X.Y. Liang	OR261469	OR261657	OR261845	OR262033
*C. hongkongensis*	CSF17122	AAAA	A	20190806-2-(18)	soil under 4-year-old *E. urophylla × E. grandis*	S.F. Chen, Q.C. Wang, L.L. Liu, Y. Liu, Y.C. Qu, Y.L. Li & X.Y. Liang	OR261470	OR261658	OR261846	OR262034
*C. hongkongensis*	CSF17125 ^b,c^	AAAA	A	20190806-2-(19)	soil under 4-year-old *E. urophylla × E. grandis*	S.F. Chen, Q.C. Wang, L.L. Liu, Y. Liu, Y.C. Qu, Y.L. Li & X.Y. Liang	OR261471	OR261659	OR261847	OR262035
*C. hongkongensis*	CSF17127	AAAA	A	20190806-2-(23)	soil under 4-year-old *E. urophylla × E. grandis*	S.F. Chen, Q.C. Wang, L.L. Liu, Y. Liu, Y.C. Qu, Y.L. Li & X.Y. Liang	OR261472	OR261660	OR261848	OR262036
*C. hongkongensis*	CSF17148	AAAA	A	20190806-2-(43)	soil under 4-year-old *E. urophylla × E. grandis*	S.F. Chen, Q.C. Wang, L.L. Liu, Y. Liu, Y.C. Qu, Y.L. Li & X.Y. Liang	OR261473	OR261661	OR261849	OR262037
*C. hongkongensis*	CSF17174	AAAA	A	20190806-2-(55)	soil under 4-year-old *E. urophylla × E. grandis*	S.F. Chen, Q.C. Wang, L.L. Liu, Y. Liu, Y.C. Qu, Y.L. Li & X.Y. Liang	OR261474	OR261662	OR261850	OR262038
*C. hongkongensis*	CSF17176 ^b^	ACAA	A	20190806-2-(56)	soil under 4-year-old *E. urophylla × E. grandis*	S.F. Chen, Q.C. Wang, L.L. Liu, Y. Liu, Y.C. Qu, Y.L. Li & X.Y. Liang	OR261475	OR261663	OR261851	OR262039
*C. hongkongensis*	CSF17178	AAAA	A	20190806-2-(57)	soil under 4-year-old *E. urophylla × E. grandis*	S.F. Chen, Q.C. Wang, L.L. Liu, Y. Liu, Y.C. Qu, Y.L. Li & X.Y. Liang	OR261476	OR261664	OR261852	OR262040
*C. pseudoreteaudii*	CSF16497	AAAA	C	20190704-3-(16)	soil under 3-year-old *E. urophylla × E. grandis*	S.F. Chen, Q.C. Wang, W.X. Wu, Y.X. Zheng & L.F. Liu	OR261477	OR261665	OR261853	OR262041
*C. pseudoreteaudii*	CSF16505 ^b,c^	AAAA	C	20190704-3-(26)	soil under 3-year-old *E. urophylla × E. grandis*	S.F. Chen, Q.C. Wang, W.X. Wu, Y.X. Zheng & L.F. Liu	OR261478	OR261666	OR261854	OR262042
*C. pseudoreteaudii*	CSF16635 ^c^	AAAA	E	20190705-1-(2)	soil under 3-year-old *E. urophylla × E. grandis*	S.F. Chen, Q.C. Wang, W.X. Wu, Y.X. Zheng & L.F. Liu	OR261479	OR261667	OR261855	OR262043
*C. pseudoreteaudii*	CSF16826 ^b,c^	AAAA	G	20190705-4-(20)	soil under 3-year-old *E. urophylla × E. grandis*	S.F. Chen, Q.C. Wang, W.X. Wu, Y.X. Zheng & L.F. Liu	OR261480	OR261668	OR261856	OR262044
*C. kyotensis*	CSF16724 ^b,c^	AAAA	F	20190705-2-(1)	soil under 5-year-old *E. urophylla × E. grandis*	S.F. Chen, Q.C. Wang, W.X. Wu, Y.X. Zheng & L.F. Liu	OR261481	OR261669	OR261857	OR262045
*C. kyotensis*	CSF16776 ^c^	AAAA	F	20190705-2-(32)	soil under 5-year-old *E. urophylla × E. grandis*	S.F. Chen, Q.C. Wang, W.X. Wu, Y.X. Zheng & L.F. Liu	OR261482	OR261670	OR261858	OR262046
*C. kyotensis*	CSF16801 ^b,c^	AAAA	F	20190705-2-(47)	soil under 5-year-old *E. urophylla × E. grandis*	S.F. Chen, Q.C. Wang, W.X. Wu, Y.X. Zheng & L.F. Liu	OR261483	OR261671	OR261859	OR262047
*C. chinensis*	CSF16829 ^b,c^	AAAA	G	20190705-4-(21)	soil under 3-year-old *E. urophylla × E. grandis*	S.F. Chen, Q.C. Wang, W.X. Wu, Y.X. Zheng & L.F. Liu	OR261484	OR261672	OR261860	OR262048

^a^ CSF: Culture collection located at the Research Institute of Fast-growing Trees (RIFT), Chinese Academy of Forestry, ZhanJiang, GuangDong Province, China. ^b^ Isolates used for phylogenetic analyses. ^c^ Isolates used for pathogenicity tests. ^d^ Genotype within each *Calonectria* species, determined by sequences of the *tef1*, *tub2*, *cmdA,* and *his3* regions. ^e^
*tef1* = translation elongation factor 1-alpha; *tub2* = β-tubulin; *cmdA* = calmodulin; *his3* = histone H3.

**Table 3 jof-09-00802-t003:** Isolates from other studies used for phylogenetic analyses in this study.

Species Code ^a^	Species	Isolate No. ^b,c^	Other Collection Number ^c^	Host	Area of Occurrence	Collector	GenBank Accession Numbers ^d^				References or Source of Data
							*cmdA*	*his3*	*tef1*	*tub2*	
B1	*Calonectria* *acaciicola*	CMW 47173^T^	CBS 143557	Soil (*Acacia auriculiformis* plantation)	Do Luong, Nghe An, Vietnam	N.Q. Pham & T.Q. Pham	MT335160	MT335399	MT412690	MT412930	[15,22]
		CMW 47174	CBS 143558	Soil (*A. auriculiformis* plantation)	Do Luong, Nghe An, Vietnam	N.Q. Pham & T.Q. Pham	MT335161	MT335400	MT412691	MT412931	[15,22]
B2	*C. acicola*	CMW 30996^T^	–	*Phoenix canariensis*	Northland, New Zealand	H. Pearson	MT335162	MT335401	MT412692	MT412932	[22,34,42]
		CBS 114812	CMW 51216	*P. canariensis*	Northland, New Zealand	H. Pearson	MT335163	MT335402	MT412693	MT412933	[22,34,42]
B4	*C. aconidialis*	CMW 35174^T^	CBS 136086; CERC 1850	Soil (*Eucalyptus* plantation)	HaiNan, China	X. Mou & S.F. Chen	MT335165	MT335404	MT412695	OK357463	[22,24,43]
		CMW 35384	CBS 136091; CERC 1886	Soil (*Eucalyptus* plantation)	HaiNan, China	X. Mou & S.F. Chen	MT335166	MT335405	MT412696	OK357464	[22,24,43]
B5	*C. aeknauliensis*	CMW 48253^T^	CBS 143559	Soil (*Eucalyptus* plantation)	Aek Nauli, North Sumatra, Indonesia	M.J. Wingfield	MT335180	MT335419	MT412710	OK357465	[15,22,24]
		CMW 48254	CBS 143560	Soil (*Eucalyptus* plantation)	Aek Nauli, North Sumatra, Indonesia	M.J. Wingfield	MT335181	MT335420	MT412711	OK357466	[15,22,24]
B8	*C. asiatica*	CBS 114073^T^	CMW 23782; CPC 3900	Debris (leaf litter)	Prathet Thai, Thailand	N.L. Hywel-Jones	AY725741	AY725658	AY725705	AY725616	[34,44]
B10	*C. australiensis*	CMW 23669^T^	CBS 112954; CPC 4714	*Ficus pleurocarpa*	Queensland, Australia	C. Pearce & B. Paulus	MT335192	MT335432	MT412723	MT412946	[22,34,45]
B17	*C. brassicicola*	CBS 112841^T^	CMW 51206; CPC 4552	Soil at *Brassica* sp.	Indonesia	M.J. Wingfield	KX784561	N/A ^e^	KX784689	KX784619	[26]
B19	*C. bumicola*	CMW 48257^T^	CBS 143575	Soil (*Eucalyptus* plantation)	Aek Nauli, North Sumatra, Indonesia	M.J. Wingfield	MT335205	MT335445	MT412736	OK357467	[15,22,24]
B20	*C. canadiana*	CMW 23673^T^	CBS 110817; STE-U 499	*Picea* sp.	Canada	S. Greifenhagen	MT335206	MT335446	MT412737	MT412958	[11,22,46,47]
		CERC 8952	–	Soil	HeNan, China	S.F. Chen	MT335290	MT335530	MT412821	MT413035	[22,32]
B23	*C. chinensis*	CMW 23674^T^	CBS 114827; CPC 4101	Soil	Hong Kong, China	E.C.Y. Liew	MT335220	MT335460	MT412751	MT412972	[22,34,44]
		CMW 30986	CBS 112744; CPC 4104	Soil	Hong Kong, China	E.C.Y. Liew	MT335221	MT335461	MT412752	MT412973	[22,34,44]
B26	*C. cochinchinensis*	CMW 49915^T^	CBS 143567	Soil (*Hevea brasiliensis* plantation)	Duong Minh Chau, Tay Ninh, Vietnam	N.Q. Pham, Q.N. Dang & T.Q. Pham	MT335225	MT335465	MT412756	MT412977	[15,22]
		CMW 47186	CBS 143568	Soil (*A. auriculiformis* plantation)	Song May, Dong Nai, Vietnam	N.Q. Pham & T.Q. Pham	MT335226	MT335466	MT412757	MT412978	[15,22]
B29	*C. colombiensis*	CMW 23676^T^	CBS 112220; CPC 723	Soil (*E. grandis* trees)	La Selva, Colombia	M.J. Wingfield	MT335228	MT335468	MT412759	MT412980	[22,44]
		CMW 30985	CBS 112221; CPC 724	Soil (*E. grandis* trees)	La Selva, Colombia	M.J. Wingfield	MT335229	MT335469	MT412760	MT412981	[22,44]
B30	*C. crousiana*	CMW 27249^T^	CBS 127198	*E. grandis*	FuJian, China	M.J. Wingfield	MT335230	MT335470	MT412761	MT412982	[22,48]
		CMW 27253	CBS 127199	*E. grandis*	FuJian, China	M.J. Wingfield	MT335231	MT335471	MT412762	MT412983	[22,48]
B31	*C. curvispora*	CMW 23693^T^	CBS 116159; CPC 765	Soil	Tamatave, Madagascar	P.W. Crous	MT335232	MT335472	MT412763	OK357468	[11,22,24,34,43,49]
		CMW 48245	CBS 143565	Soil (*Eucalyptus* plantation)	Aek Nauli, North Sumatra, Indonesia	M.J. Wingfield	MT335233	MT335473	MT412764	N/A	[15,22]
B46	*C. heveicola*	CMW 49913^T^	CBS 143570	Soil (*Hevea brasiliensis* plantation)	Bau Bang, Binh Duong, Vietnam	N.Q. Pham, Q.N. Dang & T.Q. Pham	MT335255	MT335495	MT412786	MT413004	[15,22]
		CMW 49928	CBS 143571	Soil	Bu Gia Map National Park, Binh Phuoc, Vietnam	N.Q. Pham, Q.N. Dang & T.Q. Pham	MT335280	MT335520	MT412811	MT413025	[15,22]
B48	*C. hongkongensis*	CBS 114828^T^	CMW 51217; CPC 4670	Soil	Hong Kong, China	M.J. Wingfield	MT335258	MT335498	MT412789	MT413007	[22,44]
		CERC 3570	CMW 47271	Soil (*Eucalyptus* plantation)	BeiHai, GuangXi, China	S.F. Chen, J.Q. Li & G.Q. Li	MT335260	MT335500	MT412791	MT413009	[21,22]
B51	*C. ilicicola*	CMW 30998^T^	CBS 190.50; IMI 299389; STE-U 2482	*Solanum tuberosum*	Bogor, Java, Indonesia	K.B. Boedijn & J. Reitsma	MT335266	MT335506	MT412797	OK357469	[11,22,24,34,50]
B52	*C. indonesiae*	CMW 23683^T^	CBS 112823; CPC 4508	*Syzygium aromaticum*	Warambunga, Indonesia	M.J. Wingfield	MT335267	MT335507	MT412798	MT413015	[22,44]
		CBS 112840	CMW 51205; CPC 4554	*S. aromaticum*	Warambunga, Indonesia	M.J. Wingfield	MT335268	MT335508	MT412799	MT413016	[22,44]
B55	*C. kyotensis*	CBS 114525^T^	ATCC 18834; CMW 51824; CPC 2367	*Robinia pseudoacacia*	Japan	T. Terashita	MT335271	MT335511	MT412802	MT413019	[11,22,26,51]
		CBS 114550	CMW 51825; CPC 2351	Soil	China	M.J. Wingfield	MT335246	MT335486	MT412777	MT412995	[22,26]
B57	*C. lantauensis*	CERC 3302^T^	CBS 142888; CMW 47252	Soil	LiDao, Hong Kong, China	M.J. Wingfield & S.F. Chen	MT335272	MT335512	MT412803	OK357470	[21,22,24]
		CERC 3301	CBS 142887; CMW 47251	Soil	LiDao, Hong Kong, China	M.J. Wingfield & S.F. Chen	MT335273	MT335513	MT412804	OK357471	[21,22,24]
B58	*C. lateralis*	CMW 31412^T^	CBS 136629	Soil (*Eucalyptus* plantation)	GuangXi, China	X. Zhou, G. Zhao & F. Han	MT335274	MT335514	MT412805	MT413020	[22,43]
B63	*C. lombardiana*	CMW 30602^T^	CBS 112634; CPC 4233; Lynfield 417	*Xanthorrhoea australis*	Victoria, Australia	T. Baigent	MT335395	MT335635	MT412926	MT413133	[11,22,35,45]
B66	*C. malesiana*	CMW 23687^T^	CBS 112752; CPC 4223	Soil	Northern Sumatra, Indonesia	M.J. Wingfield	MT335286	MT335526	MT412817	MT413031	[22,44]
		CBS 112710	CMW 51199; CPC 3899	Leaf litter	Prathet, Thailand	N.L. Hywel-Jones	MT335287	MT335527	MT412818	MT413032	[22,44]
B74	*C. multiseptata*	CMW 23692^T^	CBS 112682; CPC 1589	*E. grandis*	North Sumatra, Indonesia	M.J. Wingfield	MT335299	MT335539	MT412830	MT413044	[22,34,44]
B80	*C. pacifica*	CMW 16726^T^	A1568; CBS 109063; IMI 354528; STE-U 2534	*Araucaria heterophylla*	Hawaii, USA	M. Aragaki	MT335311	MT335551	MT412842	OK357472	[11,22,24,44,46]
		CMW 30988	CBS 114038	*Ipomoea aquatica*	Auckland, New Zealand	C.F. Hill	MT335312	MT335552	MT412843	OK357473	[11,22,34,44]
B86	*C. penicilloides*	CMW 23696^T^	CBS 174.55; STE-U 2388	*Prunus* sp.	Hatizyo Island, Japan	M. Ookubu	MT335338	MT335578	MT412869	MT413081	[11,22,52]
B97	*C. pseudoreteaudii*	CMW 25310^T^	CBS 123694	*E. urophylla* × *E. grandis*	GuangDong, China	M.J. Wingfield & X.D. Zhou	MT335354	MT335594	MT412885	MT413096	[22,35]
		CMW 25292	CBS 123696	*E. urophylla* × *E. grandis*	GuangDong, China	M.J. Wingfield & X.D. Zhou	MT335355	MT335595	MT412886	MT413097	[22,35]
B104	*C. queenslandica*	CMW 30604^T^	CBS 112146; CPC 3213	*E. urophylla*	Lannercost, Queensland, Australia	B. Brown	MT335367	MT335607	MT412898	MT413108	[22,35,53]
		CMW 30603	CBS 112155; CPC 3210	*E. pellita*	Lannercost, Queensland, Australia	P.Q Thu & K.M. Old	MT335368	MT335608	MT412899	MT413109	[22,35,53]
B106	*C. reteaudii*	CMW 30984^T^	CBS 112144; CPC 3201	*E. camaldulensis*	Chon Thanh, Binh Phuoc, Vietnam	M.J. Dudzinski & P.Q. Thu	MT335370	MT335610	MT412901	MT413111	[11,22,45,53]
		CMW 16738	CBS 112143; CPC 3200	*Eucalyptus* leaves	Binh Phuoc, Vietnam	M.J. Dudzinski & P.Q. Thu	MT335371	MT335611	MT412902	MT413112	[11,22,45,53]
B112	*C. sumatrensis*	CMW 23698^T^	CBS 112829; CPC 4518	Soil	Northern Sumatra, Indonesia	M.J. Wingfield	MT335382	MT335622	MT412913	OK357474	[22,24,44]
		CMW 30987	CBS 112934; CPC 4516	Soil	Northern Sumatra, Indonesia	M.J. Wingfield	MT335383	MT335623	MT412914	OK357475	[22,24,44]
B113	*C. syzygiicola*	CBS 112831^T^	CMW 51204; CPC 4511	*Syzygium aromaticum*	Sumatra, Indonesia	M.J. Wingfield	N/A	N/A	KX784736	KX784663	[26]
B116	*C. uniseptata*	CBS 413.67^T^	CMW 23678; CPC 2391; IMI 299577	*Paphiopedilum callosum*	Celle, Germany	W. Gerlach	GQ267379	GQ267248	GQ267307	GQ267208	[26]
B120	*C. yunnanensis*	CERC 5339^T^	CBS 142897; CMW 47644	Soil (*Eucalyptus* plantation)	YunNan, China	S.F. Chen & J.Q. Li	MT335396	MT335636	MT412927	MT413134	[21,22]
		CERC 5337	CBS 142895; CMW 47642	Soil (*Eucalyptus* plantation)	YunNan, China	S.F. Chen & J.Q. Li	MT335397	MT335637	MT412928	MT413135	[21,22]
B124	*C. singaporensis*	CBS 146715^T^	MUCL 048320	leaf litter (submerged in a small stream)	South East Asian rainforest, Mac Ritchie Reservoir, Singapore	C. Decock	MW890042	MW890055	MW890086	MW890124	[54]
		CBS 146713	MUCL 048171	leaf litter (submerged in a small stream)	South East Asian rainforest, Mac Ritchie Reservoir, Singapore	C. Decock	MW890040	MW890053	MW890084	MW890123	[54]
B127	*C. borneana*	CMW 50782^T^	CBS 144553	Soil (*Eucalyptus* plantation)	Sabah, Tawau, Brumas, Malaysia	N.Q. Pham, Marincowitz & M.J. Wingfield	OL635067	OL635043	OL635019	N/A	[16]
		CMW 50832	CBS 144551	Soil (*Eucalyptus* plantation)	Sabah, Tawau, Brumas, Malaysia	N.Q. Pham, Marincowitz & M.J. Wingfield	OL635065	OL635041	OL635017	N/A	[16]
B128	*C. ladang*	CMW 50776^T^	CBS 144550	Soil (*Eucalyptus* plantation)	Sabah, Tawau, Brumas, Malaysia	N.Q. Pham, Marincowitz & M.J. Wingfield	OL635075	OL635051	OL635027	N/A	[16]
		CMW 50775	CBS 144549	Soil (*Eucalyptus* plantation)	Sabah, Tawau, Brumas, Malaysia	N.Q. Pham, Marincowitz & M.J. Wingfield	OL635074	OL635050	OL635026	N/A	[16]
B129	*C. pseudomalesiana*	CMW 50821^T^	CBS 144563	Soil (*Eucalyptus* plantation)	Sabah, Tawau, Brumas, Malaysia	N.Q. Pham, Marincowitz & M.J. Wingfield	OL635076	OL635052	OL635028	OL635137	[16]
		CMW 50779	CBS 144668	Soil (*Eucalyptus* plantation)	Sabah, Tawau, Brumas, Malaysia	N.Q. Pham, Marincowitz & M.J. Wingfield	OL635077	OL635053	OL635029	OL635138	[16]
B130	*C. tanah*	CMW 50777^T^	CBS 144562	Soil (*Eucalyptus* plantation)	Sabah, Tawau, Brumas, Malaysia	N.Q. Pham, Marincowitz & M.J. Wingfield	OL635088	OL635064	OL635040	OL635146	[16]
		CMW 50771	CBS 144560	Soil (*Eucalyptus* plantation)	Sabah, Tawau, Brumas, Malaysia	N.Q. Pham, Marincowitz & M.J. Wingfield	OL635086	OL635062	OL635038	OL635144	[16]
	*C. cassiae*	ZHKUCC 210011 T	–	*Cassia surattensis*	Guangzhou CityGuangDong, China	Y. X. Zhang, C. T. Chen, Manawas., & M. M. Xiang	ON260790	N/A	MZ516860	MZ516863	[55]
		ZHKUCC 210012	–	*Cassia surattensis*	Guangzhou CityGuangDong, China	Y. X. Zhang, C. T. Chen, Manawas., & M. M. Xiang	ON260791	N/A	MZ516861	MZ516864	[55]
	*C. guangdongensis*	ZHKUCC 21-0062T	–	*Heliconia metallica*	GuangDong, China	Y. X. Zhang, C. T. Chen, Manawas., & M. M. Xiang	MZ491127	N/A	MZ491149	MZ491171	[55]
		ZHKUCC 21-0063	–	*Heliconia metallica*	GuangDong, China	Y. X. Zhang, C. T. Chen, Manawas., & M. M. Xiang	MZ491128	N/A	MZ491150	MZ491172	[55]
	*Curvicladiella cignea*	CBS 109167^T^	CPC 1595; MUCL 40269	Decaying leaf	French Guiana	C. Decock	KM231287	KM231461	KM231867	KM232002	[30,45,56]
		CBS 109168	CPC 1594; MUCL 40268	Decaying seed	French Guiana	C. Decock	KM231286	KM231460	KM231868	KM232003	[30,45,56]

^a^ Codes (B1 to B120) of the 120 accepted *Calonectria* species from [22]. ^b^ T: ex-type isolates of the species. ^c^ ATCC: American Type Culture Collection, Virginia, USA; CBS: Westerdijk Fungal Biodiversity Institute, Utrecht, The Netherlands; CERC: China Eucalypt Research Centre, ZhanJiang, GuangDong Province, China; CMW: Culture collection of the Forestry and Agricultural Biotechnology Institute (FABI), University of Pretoria, Pretoria, South Africa; CPC: Pedro Crous working collection housed at Westerdijk Fungal Biodiversity Institute; IMI: International Mycological Institute, CABI Bioscience, Egham, Bakeham Lane, UK; MUCL: Mycotheque, Laboratoire de Mycologie Systematique st Appliqee, I’Universite, Louvian-la-Neuve, Belgium; STE-U: Department of Plant Pathology, University of Stellenbosch, South Africa; ZHKUCC: Zhongkai University of Agriculture and Engineering Culture Collection; –: no other collection number. ^d^
*tef1*: translation elongation factor 1-alpha; *tub2*: *β*-tubulin; *cmdA*: calmodulin; *his3*: histone H3. ^e^ N/A: information is not available.

## Data Availability

Data are contained within the article and Appendix A.

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
