# Peer review of "Wide Distribution and Intraspecies Diversity in the Pathogenicity of *Calonectria* in Soil from *Eucalyptus* Plantations in Southern Guangxi of China"

_jof, 2023, doi:10.3390/jof9080802_

Round 1
Reviewer 1 Report
Authors have presented their research work related to the distribution and intraspecies diversity in the pathogenicity of Calonectria in Soil from Eucalyptus Plantations in GuangXi, southern China. A total of 188 Calonectria isolates were obtained from soil close to Eucalyptus trees. These isolates were identified based on DNA sequence comparisons of four partial gene regions of the translation elongation factor 1-alpha (tef1), β-tubulin (tub2), calmodulin 16 (cmdA), as well as histone H3 (his3). They have done pathogenicity tests too to satisfy the Koch postulates. Overall, it is a good effort but require some inclusions and revisions as follows:
Author citations of all scientific names are missing throughout the manuscript e.g., Eucalyptus, Calonectria, Medicago sativa…and many more
It is suggested to include sampling site map so that one can judge the distance present between sampling sites.
Discuss why some Calonectria isolates of the same species differed significantly in their virulence on the tested Eucalyptus genotypes. What could be the possible reason that the resistance of different Eucalyptus genotypes to Calonectria isolates within the same species was inconsistent.
line 31 : italicize Calonectria
table 2: accessions are missing
line 461: italicize Eucalyptus
Author Response
Our point by point responses to Reviewer One’s comments and details of the revisions have been listed and explained in the attached file “July 21, 2023_Response to Reviewer One Comments and Changes Made.docx”.

Reviewer 2 Report
Dear Authors,
Your manuscript entitled "Wide Distribution and Intraspecies Diversity in the Pathogenicity of Calonectria in Soil from Eucalyptus Plantations in GuangXi, southern China" (ID: jof-2453842) submitted to Journal of Fungi was reviewed. I thank you for writing this scientific manuscript about Calonectria spp.
Regarding to this manuscript, the following points should be considered.
Title:
The title should be written more concisely.
Abstract:
It should be noted which species had the highest virulence in the pathogenicity tests.
Introduction:
In the introduction, it is better to refer to previous studies conducted on Calonectria species on Eucalyptus plantations and other plant species in china as well as other countries (with more detailed explanations to show the importance of the species of this genus).
Materials and Methods:
1-In this part, it is necessary to consider the morphological identification of the Calonectria isolates and references used.
2-Describe the isolate considered as a positive control. Has it previously been tested for pathogenicity trial? How was it identified? on which host and genotype and …..?
Results:
1-Which morphological characteristics have been used for the identification and initial grouping of this number of isolates?
2-Table 2. takes up a lot of space in the manuscript and it might be better to consider it as Supplementary Materials and put some figures of identified species instead of this table (of course, this is only a suggestion and it is not mandatory).
3- The pathogenicity tests are a little confusing and it is better to write this section in two parts as the experiment 1 and 2. This can be more convenient for readers. This can also be considered for materials and methods.
Discussion:
The discussion seems a bit weak. It is better to use more references about the importance of species of this genus on Eucalyptus and other plant species in china as well as other countries (of course, with more explanations). Such as:
Li, J. Q., Wingfield, M. J., Liu, Q. L., Barnes, I., Roux, J., Lombard, L., Crous, P. W., and Chen, S. F. 2017. Calonectria species isolated from Eucalyptus plantations and nurseries in South China. IMA Fungus 8:259–286.
Zhang, Y.; Chen, C.; Chen, C.; Chen, J.; Xiang, M.; Wanasinghe, D.N.; Hsiang, T.; Hyde, K.D.; Manawasinghe, I.S. Identification and Characterization of Calonectria Species Associated with Plant Diseases in Southern China. J. Fungi 2022, 8, 719. https://doi.org/ 10.3390/jof8070719
Pham, N., Marincowitz, S., Chen, S. et al. Calonectria species, including four novel taxa, associated with Eucalyptus in Malaysia. Mycol Progress 21, 181–197 (2022). https://doi.org/10.1007/s11557-021-01768-8
L. Lombard, P.W. Crous, B.D. Wingfield, M.J. Wingfield, Species concepts in Calonectria (Cylindrocladium), Studies in Mycology, Volume 66, 2010, Pages 1-13, ISSN 0166-0616, https://doi.org/10.3114/sim.2010.66.01.
Dalia Aiello, Vladimiro Guarnaccia, Alessandro Vitale, Nicholas LeBlanc, Nina Shishkoff, and Giancarlo Polizzi. 2022. Impact of Calonectria Diseases on Ornamental Horticulture: Diagnosis and Control Strategies. Plant Disease, 106: 1773-1787, https://doi.org/10.1094/PDIS-11-21-2610-FE.
More recommendations:
To improve this manuscript, more suggestions and comments are included in the revised PDF file (attached file).
Best regards

I advise the respected authors to find a native English speaker to thoroughly revise the grammar of this manuscript.
Author Response
Our point by point responses to Reviewer Two’s comments and details of the revisions have been listed and explained in the attached file “July 21, 2023_Response to Reviewer Two Comments and Changes Made.docx”.
